# Random Projection Filter Bank for Time Series Data

**Amir-massoud Farahmand**
Mitsubishi Electric Research Laboratories (MERL)
Cambridge, MA, USA
`farahmand@merl.com`

**Sepideh Pourazarm**
Mitsubishi Electric Research Laboratories (MERL)
Cambridge, MA, USA
`sepid@bu.edu`

**Daniel Nikovski**
Mitsubishi Electric Research Laboratories (MERL)
Cambridge, MA, USA
`nikovski@merl.com`

## Abstract

We propose Random Projection Filter Bank (RPFB) as a generic and simple approach to extract features from time series data. RPFB is a set of randomly generated stable autoregressive filters that are convolved with the input time series to generate the features. These features can be used by any conventional machine learning algorithm for solving tasks such as time series prediction, classification with time series data, etc. Different filters in RPFB extract different aspects of the time series, and together they provide a reasonably good summary of the time series. RPFB is easy to implement, fast to compute, and parallelizable. We provide an error upper bound indicating that RPFB provides a reasonable approximation to a class of dynamical systems. The empirical results in a series of synthetic and real-world problems show that RPFB is an effective method to extract features from time series.

## 1   Introduction

This paper introduces Random Projection Filter Bank (RPFB) for feature extraction from time series data. RPFB generates a feature vector that summarizes the input time series by projecting the time series onto the span of a set of randomly generated dynamical filters. The output of RPFB can then be used as the input to any conventional estimator (e.g., ridge regression, SVM, and Random Forest [Hastie et al., 2001; Bishop, 2006; Wasserman, 2007]) to solve problems such as time series prediction, and classification and fault prediction with time series input data. RPFB is easy to implement, is fast to compute, and can be parallelized easily.

RPFB consists of a set of randomly generated filters (i.e., dynamical systems that receive inputs), which are convolved with the input time series. The filters are stable autoregressive (AR) filters, so they can capture information from the distant past of the time series. This is in contrast with more conventional approach of considering only a fixed window of the past time steps, which may not capture all relevant information. RPFB is inspired from the random projection methods [Vempala, 2004; Baraniuk and Wakin, 2009], which reduce the input dimension while preserving important properties of the data, e.g., being an approximate isometric map. It is also closely related to

Random Kitchen Sink [Rahimi and Recht, 2009] for approximating potentially infinite-dimensional reproducing kernel Hilbert space (RKHS) with a finite set of randomly selected features. RPFB can be thought of as the dynamical system (or filter) extension of these methods. RPFB is also related to the methods in the Reservoir Computing literature [Lukoševičius and Jaeger, 2009] such as Echo State Network and Liquid State Machine, in which a recurrent neural network (RNN) with random weights provides a feature vector to a trainable output layer. The difference of RPFB with them is that we are not considering an RNN as the underlying excitable dynamical system, but a set of AR filters.

The algorithmic contribution of this work is the introduction of RPFB as a generic and simple to use feature extraction method for time series data (Section 3). RPFB is a particularly suitable choice for industrial applications where the available computational power is limited, e.g., a fault prognosis system for an elevator that has only a micro-controller available. For these industrial applications, the use of powerful methods such as various adaptable RNN architectures [Hochreiter and Schmidhuber, 1997; Cho et al., 2014; Oliva et al., 2017; Goodfellow et al., 2016], which learn the feature extractor itself, might be computationally infeasible.

The theoretical contribution of this work is the finite sample analysis of RPFB for the task of time series prediction (Section 4). The theory has two main components. The first is a filter approximation error result, which provides an error guarantee on how well one might approximate a certain class of dynamical systems with a set of randomly generated filters. The second component is a statistical result providing a finite-sample guarantee for time series prediction with a generic class of linear systems. Combining these two, we obtain a finite-sample guarantee for the use of RPFB for time series prediction of a certain class of dynamical systems.

Finally, we empirically study RPFB along several standard estimators on a range of synthetic and real-world datasets (Section 5). Our synthetic data is based on Autoregressive Moving Average (ARMA) processes. This lets us closely study various aspects of the method. Moving to real-world problems, we apply RPFB to the fault diagnosis problem from ball bearing vibration measurements. We compare the performance of RPFB with that of the fixed-window history-based approach, as well as LSTM, and we obtain promising empirical results. Due to space limitation, most of the development of the theory and experimental results are reported in the supplementary material, which is an extended version of this paper. For more empirical studies, especially in the context of fault detection and prognosis, refer to Pourazarm et al. [2017].

## 2 Learning from Time Series Data

Consider a sequence $(X_1, Y_1), \ldots, (X_T, Y_T)$ of dependent random variables with $X \in \mathcal{X}$ and $Y \in \mathcal{Y}$. Depending on how $X_t$ and $Y_t$ are defined, we can describe different learning/decision making problems. For example, suppose that $Y_t = f^*(X_t) + \varepsilon_t$, in which $f^*$ is an unknown function of the current value of $X_t$ and $\varepsilon_t$ is independent of the history $X_{1:t} = (X_1, \ldots, X_t)$ and has a zero expectation, i.e., $\mathbb{E}[\varepsilon_t] = 0$. Finding an estimate $\hat{f}$ of $f^*$ using data is the standard regression (or classification) problem depending on whether $\mathcal{Y} \subset \mathbb{R}$ (regression) or $\mathcal{Y} = \{0, 1, \ldots, c - 1\}$ (classification). For example, suppose that we are given a dataset of $m$ time series $\mathcal{D}_m = \{(X_{i,1}, Y_{i,1}), \ldots, (X_{i,T_i}, Y_{i,T_i})\}_{i=1}^m$, each of which might have a varying length $T_i$. There are many methods to define an estimator for $f^*$, e.g., K-Nearest Neighbourhood, decision tree, SVM, various neural networks [Hastie et al., 2001; Bishop, 2006; Wasserman, 2007; Goodfellow et al., 2016]. An important class of estimators is based on (regularized) empirical risk minimization (ERM):

$$\hat{f} \leftarrow \underset{f \in \mathcal{F}}{\operatorname{argmin}} \frac{1}{m} \sum_{i=1}^m \frac{1}{T_i} \sum_{t=1}^{T_i} l(f(X_{i,t}), Y_{i,t}) + \lambda J(f). \tag{1}$$

Here $\mathcal{F} : \mathcal{X} \to \mathcal{Y}'$ is a function space (e.g., an RKHS with the domain $\mathcal{X}$; with $\mathcal{Y}' = \mathbb{R}$). The loss function is $l : \mathcal{Y}' \times \mathcal{Y} \to [0, \infty)$, and it determines the decision problem that is being solved, e.g., $l(y_1, y_2) = |y_1 - y_2|^2$ for the squared loss commonly used in regression. The optional regularizer (or penalizer) $J(f)$ controls the complexity of the function space, e.g., it can be $\|f\|_{\mathcal{F}}^2$ when $\mathcal{F}$ is an RKHS. The difference of this scenario with more conventional scenarios in the supervised learning and statistics is that here the input data does not satisfy the usual independence assumption anymore. Learning with dependent input data has been analyzed before [Steinwart et al., 2009; Steinwart and Christmann, 2009; Mohri and Rostamizadeh, 2010; Farahmand and Szepesvári, 2012].

More generally, however, $Y_t$ is not a function of only $X_t$, but is a function of the history $X_{1:t}$, possibly contaminated by a (conditionally) independent noise: $Y_t = f^*(X_{1:t}) + \varepsilon_t$. In the learning problem, $f^*$ is an unknown function. The special case of $f^*(X_{1:t}) = f^*(X_t)$ is the same as the previous setting.

To learn an estimator by directly using the history $X_{1:t}$ is challenging as it is a time-varying vector with an ever increasing dimension. A standard approach to deal with this issue is to use a fixed-window history-based estimator, which shall be explained next (cf. Kakade et al. [2017] for some recent theoretical results). The RPFB is an alternative approach that we describe in Section 3.

In the fixed-window history-based approach (or window-based, for short), we only look at a fixed window of the immediate past values of $X_{1:t}$. That is, we use samples in the form of $Z_t \triangleq X_{t-H+1:t}$ with a finite integer $H$ that determines the length of the window. For example, the regularized least-squares regression estimator would then be

$$\hat{f} \leftarrow \underset{f \in \mathcal{F}}{\operatorname{argmin}} \frac{1}{m} \sum_{i=1}^{m} \frac{1}{T_i - H} \sum_{t=H}^{T_i} |f(X_{i,t-H+1:t})) - Y_{i,t}|^2 + \lambda J(f), \tag{2}$$

which should be compared to (1).

A problem with this approach is that for some stochastic processes, a fixed-sized window of length $H$ is not enough to capture all information about the process. As a simple illustrative example, consider a simple moving average MA(1) univariate random process (i.e., $\mathcal{X} = \mathbb{R}$):

$$X_t = U(t) + bU(t-1) = (1 + bz^{-1})U_t, \qquad b \in (-1, 1)$$

in which $z^{-1}$ is the time-delay operator (cf. Z-transform, Oppenheim et al. 1999), i.e., $z^{-1}X_t = X_{t-1}$. Suppose that $U_t = U(t)$ $(t = 1, 2, \dots)$ is an unobservable random process that drives $X_t$. For example, it might be an independent and identically distributed (i.i.d.) Gaussian noise, which we do not observe (so it is our latent variable). To predict $Y_t = X_{t+1}$ given the previous observations $X_{1:t}$, we write $U_t = \frac{X_t}{1+bz^{-1}}$, so

$$X_{t+1} = U_{t+1} + bU_t = U_{t+1} + \frac{b}{1 + bz^{-1}}X_t = U_{t+1} + b\sum_{k \geq 0}(-b)^k X_{t-k}. \tag{3}$$

This means that $X_t$ is an autoregressive process AR($\infty$). The prediction of $X_{t+1}$ requires the value of $U_{t+1}$, which is unavailable at time $t$, and all the past values $X_{1:t}$. Since $U_{t+1}$ is unavailable, we cannot use it in our estimate, so this is the intrinsic difficulty of prediction. On the other hand, the values of $X_{1:t}$ are available to us and we can use them to predict $X_{t+1}$. But if we use a fixed-horizon window of the past values (i.e., only use $X_{t-H+1:t}$ for a finite $H \geq 1$), we would miss some information that could potentially be used. This loss of information is more prominent when the magnitude of $b$ is close to 1. This example shows that even for a simple MA(1) process with unobserved latent variables, a fixed-horizon window is not a complete summary of the stochastic process.

More generally, suppose that we have a univariate linear ARMA process

$$A(z^{-1})X_t = B(z^{-1})U_t, \tag{4}$$

with $A$ and $B$ both being polynomials in $z^{-1}$.[1] The random process $U_t$ is not available to us, and we want to design a predictor (filter) for $X_{t+1}$ based on the observed values $X_{1:t}$. Suppose that $A$ and $B$ are of degree more than 1, so we can write $A(z^{-1}) = 1 + z^{-1}A'(z^{-1})$ and $B(z^{-1}) = 1 + z^{-1}B'(z^{-1})$.[2] Assuming that $A$ and $B$ are both invertible, we use (4) to get $U_t = B^{-1}(z^{-1})A(z^{-1})X_t$. Also we can write (4) as $(1 + z^{-1}A'(z^{-1}))X_{t+1} = (1 + z^{-1}B'(z^{-1}))U_{t+1} = U_{t+1} + B'(z^{-1})U_t$. Therefore, we have

$$X_{t+1} = U_{t+1} + \left[\frac{B'(z^{-1})A(z^{-1})}{B(z^{-1})} - A'(z^{-1})\right]X_t = U_{t+1} + \frac{B'(z^{-1}) - A'(z^{-1})}{B(z^{-1})}X_t. \tag{5}$$

So if the unknown noise process $U_t$ has a zero mean (i.e., $\mathbb{E}[U_t|U_{1:t-1}] = 0$), the estimator

$$\hat{X}_{t+1}(X_{1:t}) = \frac{B'(z^{-1}) - A'(z^{-1})}{B(z^{-1})}X_t, \tag{6}$$

is unbiased, i.e., $\hat{X}_{t+1}(X_{1:t}) = \mathbb{E}[X_{t+1}|X_{1:t}]$.

If we knew the model of the dynamical system ($A$ and $B$), we could design the filter (6) to provide an unbiased prediction for the future values of $X_{t+1}$. If the learning problem is such that it requires us to know an estimate of the future observations of the dynamical system, this scheme would allow us to design such an estimator. The challenge here is that we often do not know $A$ and $B$ (or similar for other types of dynamical systems). Estimating $A$ and $B$ for a general dynamical system is a difficult task. The use of maximum likelihood-based approaches is prone to local minimum since $U$ is not known, and one has to use EM-like algorithms, cf. White et al. [2015] and references therein. Here we suggest a simple alternative based on the idea of projecting the signal onto the span of randomly generated dynamical systems. This would be RPFB, which we describe next.

## 3 Random Projection Filter Bank

The idea behind RPFB is to randomly generate many simple dynamical systems that can approximate dynamical systems such as the optimal filter in (6) with a high accuracy. Denote the linear filter in (6) as

$$\frac{B'(z^{-1}) - A'(z^{-1})}{B(z^{-1})} = \frac{p(z^{-1})}{q(z^{-1})},$$

for two polynomials $p$ and $q$, both in $z^{-1}$. Suppose that $\deg(q) = \deg(B) = d_q$ and $\deg(A) = d_A$, then $\deg(p) = d_p = \max\{d_A - 1, d_q - 1\}$. Assume that $q$ has roots $z_1, \ldots, z_{d_q} \in \mathcal{C}$ without any multiplicity. This means that

$$q(z^{-1}) = \prod_{i=1}^{d_q} (z^{-1} - z_i).$$

In complex analysis in general, and in control engineering and signal processing in particular, the roots of $q$ are known as the poles of the dynamical system and the roots of $p$ are its zeros. Any discrete-time linear time-invariant (LTI) dynamical system has such a frequency domain representation.[3]

We have two cases of either $d_p < d_q$ or $d_p \geq d_q$. We focus on the first case and describe the RPFB, and the intuition behind it. Afterwards we will discuss the second case.

**Case 1:** Suppose that $d_p < d_q$, which implies that $d_A - 1 < d_q$. We may write

$$\frac{p(z^{-1})}{q(z^{-1})} = \sum_{i=1}^{d_q} \frac{b_i}{1 - z_i z^{-1}}, \qquad (7)$$

for some choice of $b_i$s. This means that we can write (5) as

$$X_{t+1} = U_{t+1} + \frac{B'(z^{-1}) - A'(z^{-1})}{B(z^{-1})} X_t$$

$$= U_{t+1} + \sum_{i=1}^{d_q} \frac{b_i}{1 - z_i z^{-1}} X_t.$$

That is, if we knew the set of complex poles $Z_p = \{z_1, \ldots, z_{d_q}\}$ and their corresponding coefficients $B_p = \{b_1, \ldots, b_{d_q}\}$, we could provide an unbiased estimate of $X_{t+1}$ based on $X_{1:t}$. From now on, we assume that the underlying unknown system is a stable one, that is, $|z_i| \leq 1$.

Random projection filter bank is based on randomly generating many simple stable dynamical systems, which is equivalent to generating many random poles within the unit circle. Since any stable LTI filter has a representation (7) (or a similar one in Case 2), we can approximate the true dynamical system as a linear combination of randomly generated poles (i.e., filters). If the number of filters is large enough, the approximation will be accurate.

To be more precise, we cover the set of $\{z \in \mathcal{C} : |z| \leq 1\}$ with $\mathcal{N}(\varepsilon)$ random points $N_\varepsilon = \{Z'_1, \ldots, Z'_{\mathcal{N}(\varepsilon)}\}$ such that for any $z_i \in Z_p$, there exists a $Z' \in N_\varepsilon$ with $|z_i - Z'(z_i)| < \varepsilon$. Roughly

speaking, we require $\mathcal{N}(\varepsilon) = O(\varepsilon^{-2})$ random points to cover the unit circle with the resolution of $\varepsilon$. We then define the RPFB as the following set of AR filters denoted by $\phi(z^{-1})$:[4]

$$\phi(z^{-1}) : z^{-1} \mapsto \left( \frac{1}{1 - Z_1' z^{-1}}, \cdots, \frac{1}{1 - Z_{\mathcal{N}(\varepsilon)}' z^{-1}} \right). \tag{8}$$

With a slight abuse of notation, we use $\phi(X)$ to refer to the (multivariate) time series generated after passing a signal $X = (X_1, \ldots, X_t)$ through the set of filters $\phi(z^{-1})$. More concretely, this means that we convolve the signal $X$ with the impulse response of each of filters $\frac{1}{1 - Z_i' z^{-1}}$ $(i = 1, \ldots, \mathcal{N}(\varepsilon))$. Recall that the impulse response of $\frac{1}{1 - a z^{-1}}$ is the sequence $(a^t)_{t \geq 0}$, and the convolution $X * Y$ between two sequences $(X_t)_{t \geq 0}$ and $(Y_t)_{t \geq 0}$ is a new sequence

$$(X * Y)_t = \sum_\tau X_\tau Y_{t-\tau}. \tag{9}$$

We use $[\phi(X)]_i \in \mathcal{C}^{\mathcal{N}(\varepsilon)}$ to refer to the $i$-th time-step of the multivariate signal $\phi(X_{1:i})$.

The intuition of why this is a good construction is that whenever $|z_1 - z_2|$ is small, the behaviour of filter $\frac{1}{1 - z_1 z^{-1}}$ is similar to $\frac{1}{1 - z_2 z^{-1}}$. So whenever $N_\varepsilon$ provides a good coverage of the unit circle, there exists a sequence $(b_j')$ such that the dynamical system

$$\frac{p'(z^{-1})}{q'(z^{-1})} = \phi(z^{-1}) b' = \sum_{j=1}^{\mathcal{N}(\varepsilon)} \frac{b_j'}{1 - Z_j' z^{-1}}$$

behaves similar to the unknown $\frac{p}{q}$ (7). As this is a linear model, parameters $b'$ can be estimated using ordinary least-squares regression, ridge regression, Lasso, etc. For example, the ridge regression estimator for $b'$ is

$$\hat{b} \leftarrow \underset{b}{\operatorname{argmin}} \frac{1}{m} \sum_{i=1}^{m} \sum_{t=1}^{T_i} \frac{1}{T_i} \left| [\phi(X_i)]_t b - X_{i,t+1} \right|^2 + \lambda \|b\|_2^2.$$

After obtaining $\hat{b}$, we define

$$\tilde{X}(X_{1:t}; \hat{b}) = \sum_{j=1}^{\mathcal{N}(\varepsilon)} \frac{\hat{b}_j}{1 - Z_j' z^{-1}} X_{1:t},$$

which is an estimator of $\hat{X}(X_{1:t})$ (6), i.e., $\hat{X}(X_{1:t}) \approx \tilde{X}(X_{1:t}; \hat{b})$.

**Case 2:** Suppose that $d_p \geq d_q$, which implies that $d_A - 1 \geq d_q$. Then, we may write

$$\frac{p(z^{-1})}{q(z^{-1})} = R(z^{-1}) + \frac{\rho(z^{-1})}{q(z^{-1})},$$

where $\rho$ and $R$ are obtained by the Euclidean division of $p$ by $q$, i.e., $p(z^{-1}) = R(z^{-1})q(z^{-1}) + \rho(z^{-1})$ and $\deg(R) \leq d_A - 1 - d_q$ and $\deg(\rho) < d_q$. We can write:

$$\frac{p(z^{-1})}{q(z^{-1})} = \sum_{j=0}^{d_A - 1 - d_q} \nu_j z^{-j} + \sum_{i=1}^{d_q} \frac{b_i}{1 - z_i z^{-1}}. \tag{10}$$

This is similar to (7) of Case 1, with the addition of lag terms. If we knew the set of complex poles and their corresponding coefficients as well as the coefficients of the residual lag terms, $\nu_j$, we could provide an unbiased estimate of $X_{t+1}$ based on $X_{1:t}$. Since we do not know the location of poles, we randomly generate them as before. For this case, the feature set (8) should be expanded to

$$\phi(z^{-1}) : z^{-1} \mapsto \left( \left[ 1, z^{-1}, .., z^{-(d_A - 1 - d_q)} \right], \frac{1}{1 - Z_1' z^{-1}}, \cdots, \frac{1}{1 - Z_{\mathcal{N}(\varepsilon)}' z^{-1}} \right), \tag{11}$$

**Algorithm 1** Random Projection Filter Bank
---
// $\mathcal{D}_m = \{(X_{i,1}, Y_{i,1}), \ldots, (X_{i,T_i}, Y_{i,T_i})\}_{i=1}^m$: Input data
// $l : \mathcal{Y}' \times \mathcal{Y} \to \mathbb{R}$: Loss function
// $\mathcal{F}$: Function space
// $n$: Number of filters in the random projection filter bank
Draw $Z_1', \ldots, Z_n'$ uniformly random within the unit circle
Define filters $\phi(z^{-1}) = \left( \frac{1}{1 - Z_1' z^{-1}}, \ldots, \frac{1}{1 - Z_n' z^{-1}} \right)$
**for** $i = 1$ to $m$ **do**
    Pass the $i$-th time series through all the random filters $\phi(z^{-1})$, i.e., $X_{i,1:T_i}' = \phi(z^{-1}) * X_{i,1:T_i}$
**end for**
Find the estimator using extracted features $(X_{i,1:T_i}')$, e.g., by solving the regularized empirical risk minimization:

$$\hat{f} \leftarrow \operatorname*{argmin}_{f \in \mathcal{F}} \sum_{i=1}^m \sum_{t=1}^{T_i} l(f(X_{i,t}'), Y_{i,t}) + \lambda J(f). \tag{12}$$

**return** $\hat{f}$ and $\phi$

---

which consists of a history window of length $d_A - 1 - d_q$ and the random projection filters. The regressor should then estimate both $b_i$s and $\nu_i$s in (10).

RPFB is not limited to time series prediction with linear combination of filtered signals. One may use the generated features as the input to any other estimator too. RPFB can be used for other problems such as classification with time series too. Algorithm 1 shows how RPFB is used alongside a regularized empirical risk minimization algorithm. The inputs to the algorithm are the time series data $\mathcal{D}_m$, with appropriate target values created depending on the problem, the pointwise loss function $l$, the function space $\mathcal{F}$ of the hypotheses (e.g., linear, RKHS, etc.), and the number of filters $n$ in the RPFB. The first step is to create the RPFB by randomly selecting $n$ stable AR filters. We then pass each time series in the dataset through the filter bank in order to create filtered features, i.e., the feature are created by convolving the input time series with the filters' impulse responses. Finally, taking into account the problem type (regression or classification) and function space, we apply conventional machine learning algorithms to estimate $\hat{f}$. Here we present a regularized empirical risk minimizer (12) as an example, but other choices are possible too, e.g., decision trees or K-NN. We note that the use of $\phi(z^{-1}) * X_{i,1:T_i}$ in the description of the algorithm should be interpreted as the convolution of the impulse response of $\phi(z^{-1})$, which is in the time domain, with the input signal.

*Remark* 1. In practice, whenever we pick a complex pole $Z' = a + jb$ with $j = \sqrt{-1}$, we also pick its complex conjugate $\bar{Z}' = a - jb$ in order to form a single filter $\frac{1}{(1 - Z' z^{-1})(1 - \bar{Z}' z^{-1})}$. This guarantees that the output of this second-order filter is real valued.

*Remark* 2. RPFB is described for a univariate time series $X_t \in \mathbb{R}$. To deal with multivariate time series (i.e., $X_t \in \mathbb{R}^d$ with $d > 1$) we may consider each dimension separately and pass each one through RPFB. The filters in RPFB can be the same or different for each dimension. The state of the filters, of course, depends on their input, so it would be different for each dimension. If we have $n$ filters and $d$-dimensional time series, the resulting vector $X_{i,t}'$ in Algorithm 1 would be $nd$ dimensional. Randomly choosing *multivariate* filters is another possibility, which is a topic of future research.

*Remark* 3. The Statistical Recurrent Unit (SRU), recently introduced by Oliva et al. [2017], has some similarities to RPFB. SRU uses a set of exponential moving averages at various time scales to summarize a time series, which are basically AR(1) filters with real-valued poles. SRU is more complex, and potentially more expressive, than RPFB as it has several adjustable weights. On the other hand, it does not have the simplicity of RPFB. Moreover, it does not yet come with the same level of theoretical justifications as RPFB has.

## 4 Theoretical Guarantees

This section provides a finite-sample statistical guarantee for a time series predictor that uses RPFB to extract features. We specifically focus on an empirical risk minimization-based (ERM) estimator. Note that Algorithm 1 is not restricted to time series prediction problem, or the use of ERM-based

estimator, but the result of this section is. We only briefly present the results, and refer the reader to the same section in the supplementary material for more detail, including the proofs and more discussions.

Consider the time series $(X_1, X_2, \dots)$ with $X_t \in \mathcal{X} \subset [-B, B]$ for some finite $B > 0$. We denote $\mathcal{X}^* = \cup_{t \geq 1} \mathcal{X}^t$. The main object of interest in time series prediction is the conditional expectation of $X_{t+1}$ given $X_{1:t}$, which we denote by $h^*$, i.e.,[5]

$$h^*(X_{1:t}) = \mathbb{E}\left[X_{t+1} | X_{1:t}\right]. \tag{13}$$

We assume that $h^*$ belongs to the space of linear dynamical systems that has $M \in \mathbb{N}$ stable poles all with magnitude less than $1 - \varepsilon_0$ for some $\varepsilon_0 > 0$, and an $\Lambda$-bounded $\ell_p$-norm on the weights:

$$\mathcal{H}_{\varepsilon_0, M, p, \Lambda} \triangleq \left\{ \sum_{i=1}^{M} \frac{w_i}{1 - z_i z^{-1}} \ : \ |z_i| \leq 1 - \varepsilon_0, \|w\|_p \leq \Lambda \right\}. \tag{14}$$

If the value of $\varepsilon_0$, $M$, $p$, or $\Lambda$ are clear from context, we might refer to $\mathcal{H}_{\varepsilon_0, M, p, \Lambda}$ by $\mathcal{H}$. Given a function (or filter) $h \in \mathcal{H}$, here $h(x_{1:t})$ refers to the output at time $t$ of convolving a signal $x_{1:t}$ through $h$.

To define RPFB, we randomly draw $n \geq M$ independent complex numbers $Z'_1, \dots, Z'_n$ uniformly within a complex circle with radius $1 - \varepsilon_0$, i.e., $|Z'_i| \leq 1 - \varepsilon_0$ (cf. Algorithm 1). The RPFB is

$$\phi(z^{-1}) = \left( \frac{1}{1 - Z'_1 z^{-1}}, \dots, \frac{1}{1 - Z'_n z^{-1}} \right).$$

Given these random poles, we define the following approximation (filter) spaces

$$\tilde{\mathcal{H}}_{\Lambda} = \tilde{\mathcal{H}}_{n, p, \Lambda} = \left\{ \sum_{i=1}^{n} \frac{\alpha_i}{1 - Z'_i z^{-1}} \ : \ \|\alpha\|_p \leq \Lambda \right\}. \tag{15}$$

Consider that we have a sequence $(X_1, X_2, \dots, X_T, X_{T+1}, X_{T+2})$. By denoting $Y_t = X_{t+1}$, we define $((X_1, Y_1), \dots, (X_T, Y_T), (X_{T+1}, Y_{T+1}))$. We assume that $|X_t|$ is $B$-bounded almost surely. Define the estimator $\hat{h}$ by solving the following ERM:

$$\hat{h}' \leftarrow \underset{h \in \tilde{H}_{\Lambda}}{\operatorname{argmin}} \frac{1}{T} \sum_{t=1}^{T} |h(X_{1:t}) - Y_t|^2, \qquad \hat{h} \leftarrow \mathsf{Tr}_B\left[\hat{h}'\right]. \tag{16}$$

Here $\mathsf{Tr}_B\left[\hat{h}'\right]$ truncates the values of $\hat{h}'$ at the level of $\pm B$. So $\hat{h}$ belongs to the following space

$$\tilde{\mathcal{H}}_{\Lambda, B} = \left\{ \mathsf{Tr}_B\left[\tilde{h}\right] \ : \ \tilde{h} \in \tilde{\mathcal{H}}_{\Lambda} \right\}. \tag{17}$$

A central object in our result is the notion of discrepancy, introduced by [Kuznetsov and Mohri, 2015]. Discrepancy captures the non-stationarity of the process with respect to the function space.[6] of

**Definition 1** (Discrepancy—Kuznetsov and Mohri 2015). *For a stochastic process $X_1, X_2, \dots$, a function space $\mathcal{H} : \mathcal{X}^* \to \mathbb{R}$, and $T \in \mathbb{N}$, define*

$$\Delta_T(H) \triangleq \sup_{h \in \mathcal{H}} \left\{ \mathbb{E}\left[ |h(X_{1:T+1}) - Y_{T+1}|^2 | X_{1:T+1} \right] - \frac{1}{T} \sum_{t=1}^{T} \mathbb{E}\left[ |h(X_{1:t}) - Y_t|^2 | X_{1:t} \right] \right\}.$$

If the value of $T$ is clear from the context, we may use $\Delta(H)$ instead. The following is the main theoretical result of this work.

**Theorem 1.** *Consider the time series $(X_1, \ldots, X_{T+2})$, and assume that $|X_t| \leq B$ (a.s.). Without loss of generality suppose that $B \geq 1$. Let $0 < \varepsilon_0 < 1$, $M \in \mathbb{N}$, and $\Lambda > 0$ and assume that the conditional expectation $h^*(X_{1:t}) = \mathbb{E}[X_{t+1}|X_{1:t}]$ belongs to the class of linear filters $\mathcal{H}_{\varepsilon_0, M, 2, \Lambda}$ (14). Set an integer $n \geq M$ for the number of random projection filters and let $\tilde{\mathcal{H}}_\Lambda = \tilde{\mathcal{H}}_{n, 2, \Lambda}$ (15) and the truncated space be $\tilde{\mathcal{H}}_{\Lambda, B}$ (17). Consider the estimator $\hat{h}$ that is defined as (16). Without loss of generality assume that $\Lambda \geq \frac{\varepsilon_0}{B^2 \sqrt{n}}$ and $T \geq 2$. Fix $\delta > 0$. It then holds that there exists constants $c_1, c_2 > 0$ such that with probability at least $1 - \delta$, we have*

$$\left| \hat{h}(X_{1:T+1}) - h^*(X_{1:T+1}) \right|^2 \leq \frac{c_1 B^2 \Lambda}{\varepsilon_0} \log^3(T) \sqrt{\frac{n \log(1/\delta)}{T}} + \frac{c_2 B^2 \Lambda^2}{\varepsilon_0^4} \frac{\log\left(\frac{20n}{\delta}\right)}{n} + 2\Delta(\tilde{\mathcal{H}}_{\Lambda, B}).$$

The upper bounds has three terms: estimation error, filter approximation error, and the discrepancy. The $O\left(\sqrt{\frac{n}{T}}\right)$ term corresponds to the estimation error. It decreases as the length $T$ of the time series increases. As we increase the number of filters $n$, the upper bounds shows an increase of the estimation error. This is a manifestation of the effect of the input dimension on the error of the estimator. The $O(n^{-1})$ term provides an upper bound to the filter approximation error. This error decreases as we add more filters. This indicates that RPFB provides a good approximation to the space of dynamical systems $\mathcal{H}_{\varepsilon_0, M, 2, \Lambda}$ (14). Both terms show the proportional dependency on the magnitude $B$ of the random variables in the time series, and inversely proportional dependency on the minimum distance $\varepsilon_0$ of the poles to the unit circle. Intuitively, this is partly because the output of a pole becomes more sensitive to its input as it gets closer to the unit circle. Finally, the discrepancy term $\Delta(\tilde{\mathcal{H}}_{\Lambda, B})$ captures the non-stationarity of the process, and has been discussed in detail by Kuznetsov and Mohri [2015]. Understanding the conditions when the discrepancy gets close to zero is an interesting topic for future research.

By setting the number of RP filters to $n = \frac{T^{1/3} \Lambda^{2/3}}{\varepsilon_0^2}$, and under the condition that $\Lambda \leq T$, we can simplify the upper bound to

$$\left| \hat{h}(X_{1:T+1}) - h^*(X_{1:T+1}) \right|^2 \leq \frac{cB^2 \Lambda^{4/3} \log^3(T) \sqrt{\log(\frac{1}{\delta})}}{\varepsilon_0^2 T^{1/3}} + 2\Delta(\tilde{\mathcal{H}}_{\Lambda, B}),$$

which holds with probability at least $1 - \delta$, for some constant $c > 0$. As $T \to \infty$, the error converges to the level of discrepancy term.

We would like to comment that in the statistical part of the proof, instead of using the independent block technique of Yu [1994] to analyze a mixing processes [Doukhan, 1994], which is a common technique used by many prior work such as Meir [2000]; Mohri and Rostamizadeh [2009, 2010]; Farahmand and Szepesvári [2012], we rely on more recent notions of sequential complexities [Rakhlin et al., 2010, 2014] and the discrepancy [Kuznetsov and Mohri, 2015] of the function space-stochastic process couple.

This theorem is for Case 1 in Section 3, but a similar result also holds for Case 2. We also mention that as the values of $M$, $\varepsilon_0$, and $\Lambda$ of the true dynamical system space $\mathcal{H}_{\varepsilon_0, M, 2, \Lambda}$ are often unknown, the choice of number of filters $n$ in RPFB, the size of the space $M$, etc. cannot be selected based on them. Instead one should use a model selection procedure to pick the appropriate values for these parameters.

# 5 Experiments

We use a ball bearing fault detection problem to empirically study RPFB and compare it with a fixed-window history-based approach. The supplementary material provides several other experiments, including the application of LSTM to solve the very same problem, close comparison of RPFB with fixed-window history-based approach on an ARMA time series prediction problem, and a heart rate classification problem. For further empirical studies, especially in the context of fault detection and prognosis, refer to Pourazarm et al. [2017].

Reliable operation of rotating equipments (e.g., turbines) depends on the condition of their bearings, which makes the detection of whether a bearing is faulty and requires maintenance of crucial importance. We consider a bearing vibration dataset provided by Machinery Failure Prevention

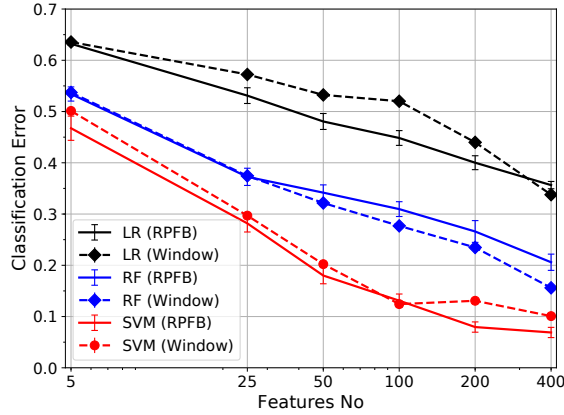

Figure 1: (Bearing Dataset) Classification error on the test dataset using RPFB and fixed-window history-based feature sets. The RPFB results are averaged over 20 independent randomly selected RPFB. The error bars show one standard error.

Technology (MFPT) Society in our experiments.[7] Fault detection of bearings is an example of industrial applications where the computational resources are limited, and fast methods are required, e.g., only a micro-controller or a cheap CPU, and not a GPU, might be available.

The dataset consists of three univariate time series corresponding to a baseline (good condition/class 0), an outer race fault (class 1), and inner race fault (class 2). The goal is to find a classifier that predicts the class label at the current time $t$ given the vibration time series $X_{1:t}$. In a real-world scenario, we train the classifier on a set of previously recorded time series, and later let it operate on a new time series observed from a device. The goal would be to predict the class label at each time step as new data arrives. Here, however, we split each of three time series to a training and testing subsets. More concretely, we first pass each time series through RPFB (or define a fixed-window of the past $H$ values of them). We then split the processed time series, which has the dimension of the number of RPFB or the size of the window, to the training and testing sets. We select the first 3333 time steps to define the training set, and the next 3333 data points as the testing dataset. As we have three classes, this makes the size of training and testing sets both equal to 10K. We process each dimension of the features to have a zero mean and a unit variance for both feature types. We perform 20 independent runs of RPFB, each of which with a new set of randomly selected filters.

Figure 1 shows the classification error of three different classifier (Logistic Regression (LR) with the $\ell_2$ regularization, Random Forest (RF), and Support Vector Machine (SVM) with Gaussian kernel) on both feature types, with varying feature sizes. We observe that as the number of features increases, the error of all classifiers decreases too. It is also noticeable that the error heavily depends on the type of classifier, with SVM being the best in the whole range of number of features. The use of RPFB instead of fixed-window history-based one generally improves the performance of LR and SVM, but not for RF. Refer to the supplementary material for more detail on the experiment.

## 6    Conclusion

This paper introduced Random Projection Filter Bank (RPFB) as a simple and effective method for feature extraction from time series data. RPFB comes with a finite-sample error upper bound guarantee for a class of linear dynamical systems. We believe that RPFB should be a part of the toolbox for time series processing.

A future research direction is to better understand other dynamical system spaces, beyond the linear one considered here, and to design other variants of RPFB beyond those that are defined by stable linear autoregressive filters. Another direction is to investigate the behaviour of the discrepancy factor.

**Acknowledgments**

We would like to thank the anonymous reviewers for their helpful feedback.

## Footnotes

[1] We assume that $A$ and $B$ both have roots within the unit circle, i.e., they are stable.

[2] The fact that both of these polynomials have a leading term of 1 does not matter in this argument.

[3]For continuous-time systems, we may use Laplace transform instead of Z-transform, and have similar representations.

[4]One could generate different types of filters, for example those with nonlinearities, but in this work we focus on linear AR filters to simplify the analysis.

[5]We use $h$ instead of $f$ to somehow emphasize that the discussion is only for the time series prediction problem, and not general estimation problem with a time series.

[6]Our definition is a simplified version of the original definition (by selecting $q_t = 1/T$ in their notation).

[7]Available from http://www.mfpt.org/faultdata/faultdata.htm.

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
