[Supplementary Material · RPFB(NIPS2017)(extended).pdf]

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

We develop tools necessary to analyze the RPFB-based time series prediction problem, i.e., predicting $\mathbb{E}[X_{t+1}|X_{1:t}]$ given $X_{1:t}$. This corresponds to the choice of $Y_t = X_{t+1}$ and the loss function $l(y_1, y_2) = |y_1 - y_2|^2$ in Algorithm 1. We further restrict the analyze to the ERM procedure, as opposed to a regularized ERM. Note that the time series prediction problem is only a subset of the possible problems that can benefit from RPFB as the feature extractor.

In Section 4.1 we develop filter approximation theory, which indicates how well a RPFB can approximate a certain class of dynamical systems. We focus on the statistical aspect of the theory in Section 4.2, and provide a guarantee for time series prediction for a large class of dynamical systems in Section 4.2.1. All these result in Theorem 8 in Section 4.3, which provides a finite-sample error upper bound guarantee for the RPFB-based time series prediction.

## 4.1    Filter Approximation Guarantees

### 4.1.1    Random Covering of $\mathbb{R}^d$

Let us denote an Euclidean ball in $\mathbb{R}^d$ centered at $x$ with radius $R > 0$ by $B_d(x, R)$. If $x = 0$ or the position of $x$ is not relevant to the argument, we simply use $B_d(R)$. Let $\mu \in \mathcal{M}(\mathbb{R}^d)$ be the uniform distribution over $B_d(R)$, and $\lambda \in \mathcal{M}(\mathbb{R}^d)$ be the Lebesgue measure, which we use to measure the volume of a ball. We have the following lemma.

**Lemma 1.** *Consider a ball $B_d(R)$ with $R \geq \frac{1}{2}$. Let $X_1, \ldots, X_n$ be a set of independent random points uniformly distributed in $B_d(R)$. For any fixed $\delta > 0$, with probability at least $1 - \delta$, it holds that*

$$\min_i \sup_{x \in B_d(R)} \|x - X_i\|_2 \leq 2R \sqrt[d]{\frac{2d \log\left(\frac{20Rn}{\delta}\right)}{n}}.$$

*Proof.* The strategy is to first cover $B_d(R)$ deterministically, and then argue that randomly choosing $\{X_i\}_{i=1}^n$ could replace that covering with high probability.

Let $X^\circ_{\varepsilon/2} = \{x^*_1, \ldots, x^*_N\}$ be an optimal $\frac{\varepsilon}{2}$-cover of $B_d(R)$, w.r.t. the Euclidean norm, with the covering number of $N = \mathcal{N}(\varepsilon/2, B_d(R), \|\cdot\|_2)$. We define the sets $A_i = B_d(x^*_i, \varepsilon/2)$ for $i = 1, \ldots, \mathcal{N}(\varepsilon/2, B_d(R), \|\cdot\|_2)$, i.e., balls located at the points in the covering set with radius $\varepsilon/2$.

Given $n$ independent random points $\{X_j\}_{j=1}^n$ uniformly distributed in $B_d(R)$, define integer-valued random variables $M_i$ to be the number of $X_j$s that fall within $A_i$, i.e.,

$$M_i = |\{X_j \in A_i : j = 1, \ldots, n\}|.$$

Now consider any $x \in B_d(R)$. By the definition of the covering set, there exists $x^*_i \in X^\circ_{\varepsilon/2}$ such that $\|x - x^*\| \leq \varepsilon/2$. Moreover, if $M_i \geq 1$, there exists a random point $X_j$ that falls within the set $A_i$. In that case, we have $\|X_j - x^*_i\| \leq \varepsilon/2$. Therefore, by triangle inequality, $\|X_j - x\| \leq \|X_j - x^*_i\| + \|x^*_i - x\| \leq \varepsilon$, which means that for that $x$, there exists a point in the random set that is less than $\varepsilon$-away from it. This may not hold, if one or more of $M_i = 0$. So the failure probability is

$$\delta \triangleq \mathbb{P}\left\{\min_i \sup_{x \in B_d(\mathbb{R}^d)} \|x - X_i\|_2 \geq \varepsilon\right\} \leq \mathbb{P}\{M_1 = 0 \vee \cdots \vee M_N = 0\}$$

$$\leq \sum_{i=1}^N \mathbb{P}\{m_i = 0\} = \sum_{i=1}^N (1 - \mu(A_i))^n$$

$$\leq \sum_{i=1}^N \exp(-n\mu(A_i)), \tag{13}$$

where the equality on the second line is because the probability that a sample from a uniformly distributed $X$ does not hit $A_i$ is $1 - \mu(A_i)$, and the last inequality is because $1 - x \leq e^{-x}$. Next, we provide the value of $\mu(A_i)$ and an upper bound on $N$.

Because $\mu$ is a uniform distribution over $B_d(R)$, the probability $\mu(A_i)$ is the ratio of the volume of $A_i$, a ball with radius $\varepsilon/2$, measured according to the Lebesgue measure, to the volume of $B_d(R)$, i.e.,

$$\mu(A_i) = \frac{\lambda(A_i)}{\lambda(B_d(R))} = \left(\frac{\varepsilon}{2R}\right)^d.$$

Lemma 9 in Appendix A.1 provides an upper bound on $N$:

$$N = \mathcal{N}\left(\frac{\varepsilon}{2}, B_d(R), \|\cdot\|_2\right) \leq \left(\frac{10R}{\varepsilon}\right)^d.$$

Therefore, we upper bound (13) by

$$\delta \leq \left(\frac{10R}{\varepsilon}\right)^d \exp\left(-\frac{n\varepsilon^d}{(2R)^d}\right) = (10R)^d \exp\left(-\left[\frac{n\varepsilon^d}{(2R)^d} - d\log\frac{1}{\varepsilon}\right]\right). \tag{14}$$

Consider the condition that

$$\frac{n\varepsilon^d}{(2R)^d} - d\log\frac{1}{\varepsilon} \geq \frac{1}{2}\frac{n\varepsilon^d}{(2R)^d}, \tag{15}$$

which holds whenever $\varepsilon \geq \sqrt[d]{\frac{2(2R)^d d\log(1/\varepsilon)}{n}}$. To simplify this, suppose for a moment that $\varepsilon \geq \frac{1}{2n}$. In this case, if we have

$$\varepsilon \geq \sqrt[d]{\frac{2(2R)^d d\log 2n}{n}}, \tag{16}$$

condition (15) holds, too. It is easy to see that for $R > 1/2$ (or any other $R > 0$ bounded away from zero, at the cost of changing constants), we also have $\varepsilon \geq \frac{1}{2n}$, as required earlier.

Under condition (16), we upper bound (14) by

$$\delta \leq (10R)^d \exp\left(-\frac{n\varepsilon^d}{2(2R)^d}\right).$$

After solving for $\varepsilon$ as a function of $\delta$, and considering (16), we get that with probability at least $1 - \delta$, it holds that

$$\varepsilon \leq 2R\sqrt[d]{\frac{2d\log\left(\frac{10R}{\delta}\right)}{n}} + 2R\sqrt[d]{\frac{2d\log 2n}{n}} \leq 4R\sqrt[d]{\frac{d\log\left(\frac{20Rn}{\delta}\right)}{n}},$$

as desired. $\square$

Note that for $d = 2$, we observe the $O(\frac{1}{\sqrt{n}})$ behaviour, as mention in Section 3.

### 4.1.2 Perturbation of Autoregressive Filters

This section presents a result that indicates how changing an AR filter's poles would affect its output. Let us first introduce some notations and definitions. Denote $h_p(t)$ as the impulse response of a first order AR filter with pole at $p = re^{jw} \in \mathbb{C}$. We use $H_p(z)$ to indicate this filter's frequency domain representation, i.e.,

$$H_p(z) = \frac{1}{1 - pz^{-1}} = \frac{z}{z - p}.$$

When $p$ is within the unit circle, the filter is stable.

Given an input $u(t)$, the output response of $h_p$ is

$$y(t) = (h_p * u)(t) = \sum_\tau u(\tau)h_p(t - \tau).$$

We are interested in knowing the difference between the output response of two filters $h_p$ and $h_{p'}$, given the same input signal $u$, as a function of the difference between $p$ and $p'$.

There are several ways to measure the error of a vector (or signal) $e = y_p - y_{p'}$. Let us first define some vector (or signal) norms, cf. Boyd and Doyle [1987]. For a vector $x = (x(1), x(2), \ldots, x(T)) = (x_1, x_2, \ldots, x_T)$ (in which $T$ can be set to $\infty$, if needed), we define

$$\|x\|_\infty = \sup_{t \geq 1} |x(t)|, \qquad \|x\|_2^2 = \sum_{t=1}^T |x(t)|^2, \qquad \|x\|_{rms}^2 = \limsup_{T \to \infty} \frac{1}{T} \sum_{t=1}^T |x(t)|^2.$$

We denote $\ell_p$ ($1 \leq p \leq \infty$) as the space of $p$-norm bounded vectors (or sequences).

The system (semi-)norms are

$$\|H\|_2 = \sup_{\|u\|_2 \neq 0} \frac{\|Hu\|_2}{\|u\|_2}, \qquad\qquad \|H\|_{rms} = \sup_{\|u\|_{rms} \neq 0} \frac{\|Hu\|_{rms}}{\|u\|_{rms}},$$

$$\|H\|_\infty = \sup_{w \in [0, 2\pi]} \left|H(e^{jw})\right|^2, \qquad\qquad \|H\|_2^2 = \frac{1}{2\pi} \int_{-\pi}^\pi \left|H(e^{jw})\right|^2 \, \mathrm{d}w.$$

The norm $\|H\|_2$ and the semi-norm $\|H\|_{rms}$ are (semi-)induced norms. It can also be shown that $\|H\|_{rms} = \|H\|_\infty$.

**Lemma 2.** *Suppose that $h_{p_1}$ and $h_{p_2}$ are both stable AR systems with $p_i = r_i e^{j\theta_i}$ with $0 \leq r_i < 1$, for $i = 1, 2$. Given an input signal $u$, let $e = (h_{p_1} - h_{p_2}) * u$. The following statements hold:*

*(a) If $u \in \ell_\infty$, we have*

$$\|e\|_\infty \leq \frac{|p_1 - p_2|}{(1 - \max\{r_1, r_2\})^2} \|u\|_\infty.$$

*(b) If $u \in \ell_2$, we have*

$$\|e\|_\infty \leq \|e\|_2 \leq \frac{|p_1 - p_2|}{(1 - r_1)(1 - r_2)} \|u\|_2.$$

*(c) If $u$ is rms-bounded,*

$$\|e\|_{rms} \leq \frac{|p_1 - p_2|}{(1 - r_1)(1 - r_2)} \|u\|_{rms}.$$

*Proof.* First recall Young's inequality for convolutions: For vectors $u \in \ell_p$ and $v \in \ell_q$ with $1 \leq p, q, r \leq \infty$ satisfying $\frac{1}{p} + \frac{1}{q} = 1 + \frac{1}{r}$, it holds that

$$\|u * v\|_r \leq \|u\|_p \|v\|_q.$$

For notational conciseness, we denote $\Delta h = h_{p_1} - h_{p_2}$, and its corresponding frequency domain representation $\Delta H = H_{p_1} - H_{p_2}$.

For case (a), we apply Young's inequality with the choice of $r = \infty$, $p = 1$, and $q = \infty$:

$$\|e\|_\infty \leq \|\Delta h\|_1 \|u\|_\infty. \tag{17}$$

To upper bound $\|\Delta h\|_1$, note that the impulse response of a first-order filter $h_p$ is the sequence $h_p(t) = (re^{jw})^t$ for $t = 0, 1, \ldots$. For two complex numbers $z_1, z_2$ and an integer $t \geq 1$, we have

$$z_1^t - z_2^t = (z_1 - z_2)\left(z_1^{t-1}z_2^0 + z_1^{t-2}z_2 + \ldots + z_1^1 z_2^{t-2} + z_1^0 z_2^{t-1}\right).$$

Setting $z_1 = p_1^t = (r_1 e^{jw_1})^t$ and $z_2 = p_2^t = (r_2 e^{jw_2})^t$, and defining $\bar{r} = \max\{r_1, r_2\}$, we obtain the upper bound

$$|p_1^t - p_2^t| \leq |p_1 - p_2| \sum_{i=0}^{t-1} |r_1 e^{jw_1}|^{t-1-i} |r_2 e^{jw_2}|^i \leq |p_1 - p_2| t\bar{r}^{t-1}.$$

Therefore,

$$\|\Delta h\|_1 = \sum_{t \geq 0} |h_{p_1}(t) - h_{p_2}(t)| \leq |p_1 - p_2| \sum_{t \geq 1} t \bar{r}^{t-1} = \frac{|p_1 - p_2|}{(1 - \bar{r})^2}.$$

To prove cases (b) and (c), we have [Boyd and Doyle, 1987]

$$\begin{aligned} \|e\|_2 &\leq \|\Delta H\|_\infty \|u\|_2, \\ \|e\|_{\text{rms}} &\leq \|\Delta H\|_\infty \|u\|_{\text{rms}}. \end{aligned} \tag{18}$$

So we need to upper bound $\|\Delta H\|_\infty$. We have

$$\left|\Delta H(e^{jw})\right|^2 = \left|\frac{e^{jw}(p_1 - p_2)}{(e^{jw} - p_1)(e^{jw} - p_2)}\right|^2 = \frac{|p_1 - p_2|^2}{|e^{jw} - p_1|^2 |e^{jw} - p_2|^2}.$$

Therefore,

$$\begin{aligned} \|\Delta H\|_\infty^2 = \max_w \left|\Delta H(e^{jw})\right|^2 &= \frac{|p_1 - p_2|^2}{\min_w \left\{|e^{jw} - p_1|^2 |e^{jw} - p_2|^2\right\}} \\ &\leq \frac{|p_1 - p_2|^2}{\min_w \left\{|e^{jw} - p_1|^2\right\} \min_w \left\{|e^{jw} - p_2|^2\right\}} \\ &= \frac{|p_1 - p_2|^2}{(1 - r_1)^2(1 - r_2)^2}. \end{aligned} \tag{19}$$

The last equality is because for $p = re^{j\theta}$ with $0 \leq r < 1$ (a point within a unit circle), we have

$$\min_w |e^{jw} - p| = \min_w |1e^{jw} - re^{j\theta}| = |1 - r|,$$

with the minimizer being $w = \theta$. By (18) and (19), we can relate $\|e\|_2$ ($\|e\|_{\text{rms}}$) to $\|u\|_2$ ($\|u\|_{\text{rms}}$) and the location of the poles $p$ and $p'$, as desired. This finishes the proof of upper bounding $\|e\|_2$ and $\|e\|_{\text{rms}}$. Noticing that $\|e\|_\infty \leq \|e\|_2$ concludes the proof. $\square$

*Remark* 4. We could prove $\|e\|_\infty \leq \frac{|p_1 - p_2|}{(1 - r_1)(1 - r_2)} \|u\|_2$ differently. By Young's inequality (17) with $r = \infty$ and $p = q = 2$, we get

$$\|e\|_\infty \leq \|\Delta h\|_2 \|u\|_2.$$

By the Parseval's theorem, which shows that $\|\Delta h(t)\|_2 = \|\Delta H(z)\|_2$, we have

$$\|e\|_\infty \leq \|\Delta H\|_2 \|u\|_2. \tag{20}$$

We need to upper bound $\|\Delta H\|_2$. For a contour $\Gamma$ in $\mathbb{C}$ with length $L(\Gamma)$ and a continuous complex-valued function $f$, it holds that

$$\left|\oint_\Gamma f(z)\mathrm{d}z\right| \leq L(\Gamma) \max_{z \in \Gamma} |f(z)|.$$

Therefore, for the contour $\Gamma$ being the unit circle (i.e., $|z| = 1$), we have

$$\begin{aligned} \|\Delta H\|_2^2 = \frac{1}{2\pi} \int_{-\pi}^{\pi} \left|\Delta H(e^{jw})\right|^2 \mathrm{d}w &= \frac{1}{2\pi} \oint_\Gamma \frac{|p_1 - p_2|^2}{|z - p_1|^2 |z - p_2|^2} \mathrm{d}z \\ &\leq \frac{2\pi}{2\pi} |p_1 - p_2|^2 \max_{z \in \Gamma} \frac{1}{|z - p_1|^2 |z - p_2|^2} \leq \frac{|p_1 - p_2|^2}{(1 - r_1)^2(1 - r_2)^2}. \end{aligned}$$

This inequality together with (20) lead to the desired result.

The following is an immediate corollary of this lemma.

**Corollary 3.** *Consider the same definitions and conditions as in Lemma 2. In addition, assume that $r_1, r_2, \leq 1 - \varepsilon_0$ for some $\varepsilon_0 > 0$ bounded away from zero. Under the condition that $u \in \ell_q$ for $q \in \{\infty, 2, rms\}$, we have*

$$\|e\|_q \leq \frac{|p_1 - p_2|}{\varepsilon_0^2} \|u\|_q.$$

This result is somehow conservative as it assumes that both $r_1$ and $r_2$ in Lemma 2 take value of $1 - \varepsilon_0$. This leads to the observed $O(\varepsilon_0^{-2})$ behaviour. Notice than if one of them, say $r_1$, takes $1 - \varepsilon_0$, but the other is well within the unit circle, the behaviour would be $O(\varepsilon_0^{-1})$. Nonetheless to simplify the rest of analysis, we use this corollary as is.

### 4.1.3 Filter Approximation Error

This section provides the main theoretical result on the filter approximation error. We show that a class of filter spaces can be approximated by a set of randomly selected filters.

Let us first introduce a class of filters (or dynamical systems) for which we will provide an approximation guarantee. Let $h_\theta$ be a filter, parametrized by $\theta \in \Theta$, that maps a sequence $x_{1:t} \subset \mathcal{X}$ to another sequence $y_{1:t} \subset \mathcal{Y}$, for any $t = 1, 2, \ldots$. We consider that $\Theta$ is a metric space, so that the distance between two points in $\Theta$ is well-defined. We denote the output of a filter parameterized by $\theta$ by $y_{1:t}^{(\theta)}$.

An example of $h_\theta$ would be a stable linear AR(1) filter $h_\theta(z) = \frac{1}{1 - \theta z^{-1}}$ where $\theta$ is within the unit circle. For such a linear system, $y(t) = (h_\theta * x)(t) = \sum_\tau h_\theta(\tau) x(t - \tau)$, cf., (9). RPFB consists of such filters. Another example is a mapping defined by an RNN, such as LSTM, which is parameterized by $\theta$.

Denote $g \in \mathcal{G}$ as a mapping from a sequence $y_{1:t}$ to a real/complex number. Therefore, $g(h_\theta(x_{1:t}))$ is a mapping from a sequence to a number. The function $g$ acts as a link function. Some examples are $g(y_{1:t}) = y_t$ (selecting the last elements of a sequence), $g(y_{1:t}) = \max_{1 \le i \le t} y_i$ (max pooling over the temporal dimension), and $g(y_{1:t}) = \frac{1}{t} \sum_{i=1}^{t} y_i$ (mean pooling over the temporal dimension).

Consider an integer number $n \in \mathbb{N}$. For a set of functions $\underline{g} = (g_1, \ldots, g_n)$ with each $g_i$ being a member of $\mathcal{G}$, a set of parameters $\underline{\theta} = (\theta_1, \ldots, \theta_n)$ with each $\theta_i \in \Theta$, and a set of weights $w \in \mathbb{R}^n$, we define a linear combination of filters on the input $x_{1:t}$ (for any $t = 1, 2, \ldots$)

$$f(x_{1:t}; \underline{w}, \underline{g}, \underline{\theta}) = \sum_{i=1}^{n} w_i g_i \left( h_{\theta_i}(x_{1:t}) \right).$$

Fix $M \in \mathbb{N}$, $1 \le p \le \infty$, and $\Lambda > 0$. We define the following classes of filters:

$$\mathcal{F}_{(M, \mathcal{G}, \Theta, p)} = \left\{ x_{1:t} \mapsto f(x_{1:t}; \underline{w}, \underline{g}, \underline{\theta}) \; : \; \underline{w} \in \mathbb{R}^M, \|\underline{w}\|_p < \infty, g_i \in \mathcal{G}, \theta_i \in \Theta, i = 1, \ldots, M \right\},$$

$$\mathcal{F}_{(M, \mathcal{G}, \Theta, p, \Lambda)} = \left\{ x_{1:t} \mapsto f(x_{1:t}; \underline{w}, \underline{g}, \underline{\theta}) \; : \; \underline{w} \in \mathbb{R}^M, \|\underline{w}\|_p \le \Lambda, g_i \in \mathcal{G}, \theta_i \in \Theta, i = 1, \ldots, M \right\}.$$
(21)

We use $\mathcal{F}$ when parameters $M$, $\mathcal{G}$, $\Theta$, $p$, and $\Lambda$ are clear from context. Occasionally we may only use those subscripts that are relevant to the context, e.g., $\mathcal{F}_\Lambda$ is $\mathcal{F}_{(M, \mathcal{G}, \Theta, p, \Lambda)}$ for certain parameters $M$, $\mathcal{G}$, $\Theta$, and $p$ that should be clear from the context.

The true filter, which we want to approximate, is $f^*(x_{1:t}) = f(x_{1:t}; \underline{w}^*, \underline{g}^*, \underline{\theta}^*) \in \mathcal{F}$ for a certain unknown $\underline{w}^*$, $\underline{g}^*$, and $\underline{\theta}^*$.

Now consider an ordered set of parameters $\tilde{\Theta} = \{\tilde{\theta}_1, \ldots, \tilde{\theta}_n\}$ with $\tilde{\theta}_i \in \Theta$. We shall later choose these $\tilde{\theta}$ to be random parameters that define a random projection filter bank. For the parameter space $\tilde{\Theta}$, we can define an index function $I(\theta) : \theta \mapsto \{1, \ldots, n\}$ that returns the index of a member of $\tilde{\Theta}$ that is closest to $\theta$. Given $\tilde{\Theta}$ and for $n \ge M$, we define the filter space constructed by $\tilde{\Theta}$:

$$\tilde{\mathcal{F}} = \mathcal{F}_{(n, \mathcal{G}, \tilde{\Theta}, p)},$$
$$\tilde{\mathcal{F}}_\Lambda = \mathcal{F}_{(n, \mathcal{G}, \tilde{\Theta}, p, \Lambda)}.$$
(22)

We make two assumptions. We will later show the condition that they hold (cf. Proposition 5 for Assumption A2 and the proof of Theorem 6 for Assumption A1).

**Assumption A1** For a given $1 \le q, r \le \infty$, and for any sequence $(x_t) \in \ell_r$, there exists a finite $\varepsilon < \infty$ such that for any $\theta \in \Theta$ and $t = 1, 2, \ldots$, we have

$$\left\| y_{1:t}^{(\theta)} - y_{1:t}^{(\tilde{\theta}_{I(\theta)})} \right\|_q \le \varepsilon \|x_{1:t}\|_r,$$

where $y_{1:t}^{(\theta)}$ and $y_{1:t}^{(\tilde{\theta}_{I(\theta)})}$ are the output of filters parameterized by $\theta$ and $\tilde{\theta}_{I(\theta)}$ with the same input of $x_t$.

The intuition behind this assumption is that it requires that the response $y_{1:t}$ of a filter parametrized by $\theta$ is similar to the response of its closest filter within the filter space defined by $\tilde{\Theta}$. The error can depend on the size of the input signal $x_{1:t}$. In the proof of Theorem 6 we show that this assumption holds for RPFB with a certain choice of $\varepsilon$, $q$, and $r$.

**Assumption A2** Fix $1 \leq p, q \leq \infty$. For any $g \in \mathcal{G}$ and for any $T \in \mathbb{N}$ and for any $y_{1:T}$ and $y'_{1:T}$, there exists $L < \infty$ such that

$$\sqrt[p]{\sum_{t=1}^{T} |g(y_{1:t}) - g(y'_{1:t})|^p} \leq L_{q \to p} \|y_{1:T} - y'_{1:T}\|_q, \qquad (1 \leq p < \infty)$$

$$\max_{1 \leq t \leq T} |g(y_{1:t}) - g(y'_{1:t})| \leq L_{q \to \infty} \|y_{1:T} - y'_{1:T}\|_q. \qquad (p = \infty)$$

The intuition behind this assumption is that it requires that the link function $g$ does not amplify its input signal too much. The amplification is characterized by $L_{q \to p}$. Proposition 5 provides some example link function for which this assumption holds. We may refer to such a $\mathcal{G}$ as an $L_{q \to p}$-Lipschitz w.r.t. $(\ell_p, \ell_q)$ link space.

Next we show that under the aforementioned assumptions, any function in $\mathcal{F}$ (21), can be approximated by a member $\tilde{\mathcal{F}}$ (22). This approximation result indicates that, under certain assumptions, a finite cover $\tilde{\Theta}$ is enough to approximate a function with parameters in $\Theta$, which might be continuous.

**Lemma 4.** *Let $1 \leq p, q, r \leq \infty$, and consider the function spaces $\mathcal{F} = \mathcal{F}_{(M,\mathcal{G},\Theta,p)}$ and $\tilde{\mathcal{F}} = \mathcal{F}_{(n,\mathcal{G},\tilde{\Theta},p)}$ as defined above. Suppose that Assumptions A1 and A2 hold. Then for any $f^* \in \mathcal{F}$, $T \in \mathbb{N}$, and input signal $x_{1:T} \in \ell_r$ we have*

$$\min_{\tilde{f} \in \tilde{\mathcal{F}}} \sqrt[p]{\sum_{t=1}^{T} \left| f^*(x_{1:t}) - \tilde{f}(x_{1:t}) \right|^p} \leq \varepsilon L_{q \to p} \|\underline{w}^*\|_p \|x_{1:T}\|_r,$$

$$\min_{\tilde{f} \in \tilde{\mathcal{F}}} \sqrt[p]{\max_{1 \leq t \leq T} \left| f^*(x_{1:t}) - \tilde{f}(x_{1:t}) \right|^p} \leq \varepsilon L_{q \to \infty} \|\underline{w}^*\|_p \|x_{1:T}\|_r.$$

*Furthermore, let $\Lambda > 0$. If $\mathcal{F} = \mathcal{F}_{(M,\mathcal{G},\Theta,p,\Lambda)}$ and $\tilde{\mathcal{F}} = \mathcal{F}_{(n,\mathcal{G},\tilde{\Theta},p,\Lambda)}$, the same inequalities hold.*

*Proof.* Consider $f^*(x_{1:t}) = f(x_{1:t}; \underline{w}^*, \underline{g}^*, \underline{\theta}^*) \in \mathcal{F}$. We define $\tilde{\underline{\theta}}^* = (\tilde{\theta}_1^*, \ldots, \tilde{\theta}_n^*)$ by

$$\tilde{\underline{\theta}}_i^* = \begin{cases} \tilde{\theta}_{I(\theta_i^*)} & 1 \leq i \leq M \\ 0 & M < i \leq n \end{cases}$$

So each element of $\tilde{\underline{\theta}}^*$ is selected to be a member of $\tilde{\Theta}$ that is the closest to $\theta_i^*$. We let $\tilde{\underline{w}} = (\underline{w}^*; \mathbf{0}_{n-M})$ and $\tilde{\underline{g}} = (\underline{g}^*; \mathbf{0}_{n-M})$, which means that the first $n$ elements of $\tilde{\underline{w}}$ (or $\tilde{\underline{g}}$) are the same as $\underline{w}^*$ (or $\underline{g}^*$), and their last $n - M$ elements are zero scalars (or functions). With these choices, we define $\tilde{f}'(x_{1:t}) = f(x_{1:t}; \tilde{\underline{w}}, \tilde{\underline{g}}, \tilde{\underline{\theta}}^*)$, which is a member of $\tilde{\mathcal{F}}$.

We have the following chain of inequalities:

$$\min_{\tilde{f}\in\tilde{\mathcal{F}}} \sum_{t=1}^{T}\left|f^*(x_{1:t}) - \tilde{f}(x_{1:t})\right|^p \overset{(a)}{\leq} \sum_{t=1}^{T}\left|f^*(x_{1:t}) - \tilde{f}'(x_{1:t})\right|^p$$

$$= \sum_{t=1}^{T}\left|\sum_{i=1}^{M} w_i^* \left[ g_i^*\left(\underbrace{h_{\theta_i^*}(x_{1:t})}_{\triangleq y_{1:t}^{(i)}}\right) - g_i^*\left(\underbrace{h_{\tilde{\theta}_{I(\theta_i^*)}^*}(x_{1:t})}_{\triangleq \tilde{y}_{1:t}^{(i)}}\right)\right]\right|^p$$

$$\overset{(b)}{\leq} \sum_{t=1}^{T}\sum_{i=1}^{M} |w_i^*|^p \left|g\left(y_{1:t}^{(i)}\right) - g\left(\tilde{y}_{1:t}^{(i)}\right)\right|^p$$

$$\overset{(c)}{=} \sum_{i=1}^{M} |w_i^*|^p \sum_{t=1}^{T} \left|g\left(y_{1:t}^{(i)}\right) - g\left(\tilde{y}_{1:t}^{(i)}\right)\right|^p$$

$$\overset{(d)}{\leq} L_{q\to p}^p \sum_{i=1}^{M} |w_i^*|^p \left\|y_{1:T}^{(i)} - \tilde{y}_{1:T}^{(i)}\right\|_q^p$$

$$\leq L_{q\to p}^p \|\underline{w}^*\|_p^p \max_{i=1,\dots,M} \left\|y_{1:T}^{(i)} - \tilde{y}_{1:T}^{(i)}\right\|_q^p$$

$$\overset{(e)}{\leq} L_{q\to p}^p \|\underline{w}^*\|_p^p \|x_{1:T}\|_r^p \varepsilon^p.$$

The inequality (a) is because of the optimizer property of $\tilde{f}$ and the fact that the function $\tilde{f}' \in \tilde{\mathcal{F}}$, as constructed above, cannot make the objective smaller than the minimizer. Jensen's inequality shows inequality (b). We use Tonelli's theorem to exchange the order of summations in the equality (c). Assumption A2 shows (d); and (e) is by Assumption A1. The definition $y_{1:t}^{(i)}$ is a short-hand for $y_{1:t}^{(\theta_i^*)}$, and likewise for $\tilde{y}_{1:t}^{(i)}$.

The proof of the other case, the supremum over the sequence, is similar too:

$$\min_{\tilde{f}\in\tilde{\mathcal{F}}} \max_{1\leq t\leq T}\left|f^*(x_{1:t}) - \tilde{f}(x_{1:t})\right|^p \leq \max_{1\leq t\leq T}\left|f^*(x_{1:t}) - \tilde{f}'(x_{1:t})\right|^p$$

$$= \max_{1\leq t\leq T}\left|\sum_{i=1}^{M} w_i^* \left[ g\left(y_{1:t}^{(i)}\right) - g\left(\tilde{y}_{1:t}^{(i)}\right)\right]\right|^p$$

$$\leq \max_{1\leq t\leq T}\sum_{i=1}^{M} |w_i^*|^p \left|g\left(y_{1:t}^{(i)}\right) - g\left(\tilde{y}_{1:t}^{(i)}\right)\right|^p$$

$$= \sum_{i=1}^{M} |w_i^*|^p \max_{1\leq t\leq T}\left|g\left(y_{1:t}^{(i)}\right) - g\left(\tilde{y}_{1:t}^{(i)}\right)\right|^p$$

$$\leq L_{q\to\infty}^p \sum_{i=1}^{M} |w_i^*|^p \left\|y_{1:T}^{(i)} - \tilde{y}_{1:T}^{(i)}\right\|_q^p$$

$$\leq L_{q\to\infty}^p \|\underline{w}^*\|_p^p \max_{i=1,\dots,M} \left\|y_{1:T}^{(i)} - \tilde{y}_{1:T}^{(i)}\right\|_q^p \leq L_{q\to\infty}^p \|\underline{w}^*\|_p^p \|x_{1:T}\|_r^p \varepsilon^p.$$

Finally note that by construction $\|\underline{\tilde{w}}\|_p = \|\underline{w}^*\|_p$ for any $p \geq 1$. Therefore, if $f^* \in \mathcal{F}_\Lambda$, no extra error is introduced by restricting the minimization to $\tilde{\mathcal{F}}_\Lambda$.[5]

$$\square$$

We now provide some examples of $\mathcal{G} = \{\, g : y_{1:t} \mapsto \mathbb{R} \,:\, t = 1, 2, \dots \}$ for which Assumption A2 holds. We only provide examples when $\mathcal{G}$ has a single function $g$, which is specified by the next proposition.

**Proposition 5.** *Consider Assumption A2 with the choice of $1 \le p, q \le \infty$, $q \ge p$ and $T \in \mathbb{N}$.*

- *If $g(y_{1:t}) = y_t$, we have $L_{q \to p} = T^{\frac{1}{p} - \frac{1}{q}}$.*

- *If $g(y_{1:t}) = \max_{1 \le i \le t} y_i$, we have $L_{\infty \to p} = T^{\frac{1}{p}}$.*

- *If $g(y_{1:t}) = \frac{1}{t} \sum_{i=1}^{t} y_i$, for $1 \le p < \infty$, we have $L_{q \to p} = \sqrt[p]{H_{p + \frac{p}{q} - 1}(T)}$, where $H_s(T)$ is the generalized Harmonic number, i.e., $H_s(T) = \sum_{t=1}^{T} \frac{1}{t^s}$. We also have $L_{\infty \to \infty} = 1$.*

*Proof.* **Case $g(y_{1:t}) = y_t$.** Given $p, q$, let $r = q/p$ and $s = q/(q - p)$. The pair $(r, s)$ satisfies $1/r + 1/s = 1$. By the application of the Hölder's inequality, we can write

$$\sum_{t=1}^{T} |g(y_{1:t}) - g(y'_{1:t})|^p = \sum_{t=1}^{T} |y_t - y'_t|^p \le \left[ \sum_{t=1}^{T} \left( |y_t - y'_t|^p \right)^r \right]^{1/r} \left[ \sum_{t=1}^{T} 1^s \right]^{1/s}$$

$$= \left[ \sum_{t=1}^{T} |y_t - y'_t|^q \right]^{p/q} T^{\frac{q-p}{q}}.$$

Raising two sides to the power of $1/p$ leads to the desired result. When $q = \infty$, we have

$$\sum_{t=1}^{T} |y_t - y'_t|^p \le T \max_{1 \le t \le T} |y_t - y'_t|^p = T \, \|y_{1:T} - y'_{1:T}\|_\infty^p.$$

**Case $g(y_{1:t}) = \max_{1 \le i \le t} y_i$.** When $p < \infty$, we have

$$\sum_{t=1}^{T} |g(y_{1:t}) - g(y'_{1:t})|^p = \sum_{t=1}^{T} \left| \max_{1 \le i \le t} y_i - \max_{1 \le i \le t} y'_i \right|^p \le \sum_{t=1}^{T} \max_{1 \le i \le t} |y_i - y'_i|^p$$

$$\le T \, \|y_{1:T} - y'_{1:T}\|_\infty^p,$$

which shows the desired result. For $p = \infty$, we have

$$\max_{1 \le t \le T} |g(y_{1:t}) - g(y'_{1:t})| \le \max_{1 \le t \le T} \max_{1 \le i \le t} |y_i - y'_i| \le \|y_{1:T} - y'_{1:T}\|_\infty.$$

**Case $g(y_{1:t}) = \frac{1}{t} \sum_{i=1}^{t} y_i$.** For the clarity of the proof, first consider $q < \infty$. We choose the same pair $(r, s)$ as in the first case. We have

$$\sum_{t=1}^{T} |g(y_{1:t}) - g(y'_{1:t})|^p = \sum_{t=1}^{T} \left| \frac{1}{t} \sum_{i=1}^{t} (y_i - y'_i) \right|^p \le \sum_{t=1}^{T} \frac{1}{t^p} \sum_{i=1}^{t} |y_i - y'_i|^p$$

$$\le \sum_{t=1}^{T} \frac{1}{t^p} \left[ \left( \sum_{i=1}^{t} |y_i - y'_i|^{pr} \right)^{1/r} \left( \sum_{i=1}^{t} 1^s \right)^{1/s} \right]$$

$$= \sum_{t=1}^{T} \frac{1}{t^{p - \frac{1}{s}}} \left( \sum_{i=1}^{t} |y_i - y'_i|^q \right)^{p/q}$$

$$\le \|y_{1:T} - y'_{1:T}\|_q^p \sum_{t=1}^{T} \frac{1}{t^{\frac{pq+p-q}{q}}}.$$

So we may choose $L_{q \to p}^p = \sum_{t=1}^{T} \frac{1}{t^{\frac{pq+p-q}{q}}} = H_{p + \frac{p}{q} - 1}(T)$.

When $q = \infty$ and $p < \infty$, we have

$$\sum_{t=1}^{T} \frac{1}{t^p} \sum_{i=1}^{t} |y_i - y'_i|^p \le \sum_{t=1}^{T} \frac{1}{t^p} t \, \|y_{1:t} - y'_{1:t}\|_\infty^p \le \|y_{1:T} - y'_{1:T}\|_\infty^p \sum_{t=1}^{T} \frac{1}{t^{p-1}},$$

so we can choose $L_{\infty \to p} = \sqrt[p]{H_{p-1}(T)}$.

When $p = q = \infty$, we have

$$\max_{1 \leq t \leq T} \left| \frac{1}{t} \sum_{i=1}^{t} (y_i - y_i') \right| \leq \max_{1 \leq t \leq T} \frac{1}{t} t \, \|y_{1:t} - y_{1:t}'\|_\infty \leq \|y_{1:T} - y_{1:T}'\|_\infty \,,$$

so we can choose $L_{\infty \to \infty} = 1$.

$\square$

As some examples of the behaviour of the (generalized) Harmonic function, we mention that $\lim_{T \to \infty} H_1(T) - \ln T = \gamma$, where $\gamma \approx 0.57721$ is the Euler-Mascheroni constant. For finite $T$, a simple upper bound is $H_1(T) \leq 2 \ln(T+1)$, which is within a factor of 2 of the optimal value. Another example is $H_2(T) \leq H_2(\infty) = \frac{\pi^2}{6}$.

### 4.1.4 Linear Filter Approximation

We consider the following space of linear dynamical systems that has $M \in \mathbb{N}$ stable poles all with magnitude less than or equal to $1 - \varepsilon_0$ for some $\varepsilon_0 > 0$ and a bounded $\ell_p$-norm on the weights (with the bound of $\Lambda > 0$ in the second definition):

$$\mathcal{H}_{\varepsilon_0, M, p} \triangleq \left\{ \sum_{i=1}^{M} \frac{w_i}{1 - z_i z^{-1}} \; : \; |z_i| \leq 1 - \varepsilon_0, \|w\|_p < \infty \right\},$$

$$\mathcal{H}_{\varepsilon_0, M, p, \Lambda} \triangleq \left\{ \sum_{i=1}^{M} \frac{w_i}{1 - z_i z^{-1}} \; : \; |z_i| \leq 1 - \varepsilon_0, \|w\|_p \leq \Lambda \right\}. \tag{23}$$

If the values of $\varepsilon_0$, $M$, $p$, or $\Lambda$ are clear from context, we might refer to $\mathcal{H}_{\varepsilon_0, M, p}$ by $\mathcal{H}$ (and likewise for $\mathcal{H}_{\varepsilon_0, M, p, \Lambda}$ by $\mathcal{H}_\Lambda$). A function $h \in \mathcal{H}$ is identified by its set of $\{z_i\}_{i=1}^{M}$ and the vector $w = (w_1, \ldots, w_M)$. We may switch back and forth between these two representations.

Given a function (or filter) $h \in \mathcal{H}$, we use $h(x_{1:t})$ as a shorthand to refer to the output at time $t$ of convolving a signal $x_{1:t}$ through $h$. In the notation of Section 4.1.3, this refers to $g(h(x_{1:t}))$ with $g(y_{1:t}) = y_t$.

To define the random projection filter bank, we randomly draw $n \geq M$ independent complex numbers $Z_1', \ldots, Z_n'$ uniformly from a complex circle with radius $1 - \varepsilon_0$, i.e., $|Z_i| \leq 1 - \varepsilon_0$ (cf. Algorithm 1). The RPFB is

$$\phi(z^{-1}) = \left( \frac{1}{1 - Z_1' z^{-1}}, \ldots, \frac{1}{1 - Z_n' z^{-1}} \right).$$

Given these random poles, we define the following approximation (filter) spaces:

$$\tilde{\mathcal{H}}_{n,p} = \left\{ \sum_{i=1}^{n} \frac{\alpha_i}{1 - Z_i' z^{-1}} \; : \; \|\alpha\|_p < \infty \right\}, \tilde{\mathcal{H}}_{n,p,\Lambda} = \left\{ \sum_{i=1}^{n} \frac{\alpha_i}{1 - Z_i' z^{-1}} \; : \; \|\alpha\|_p \leq \Lambda \right\}. \tag{24}$$

These define randomly constructed dynamical system spaces, which as we show, approximate $\mathcal{H}_{\varepsilon_0, M, p}$ and $\mathcal{H}_{\varepsilon_0, M, p, \Lambda}$ (23). We are now ready to state the main result of this section.

**Theorem 6.** *Consider the class of LTI systems $\mathcal{H}_{\varepsilon_0, M, p}$ for $0 < \varepsilon_0 < 1$, $M \geq 1$, and $p \in \{1, 2\}$. For $n \geq M$, define the space of RPFB as described above. Fix $\delta > 0$. For any input signal $x_{1:T} \in \ell_2$ ($T \in \mathbb{N}$), the following statements hold with probability at least $1 - \delta$:*

- *For any dynamical system $h^* \in \mathcal{H}_{\varepsilon_0, M, 1}$, with its corresponding $w^*$, we have*

$$\min_{\tilde{h} \in \tilde{\mathcal{H}}_{n,1}} \sum_{t=1}^{T} \left| h^*(x_{1:t}) - \tilde{h}(x_{1:t}) \right| \leq \frac{4}{\varepsilon_0^2} \sqrt{\frac{T \log(\frac{20n}{\delta})}{n}} \, \|w^*\|_1 \, \|x_{1:T}\|_2 \,.$$

- *For any dynamical system $h^* \in \mathcal{H}_{\varepsilon_0, M, 2}$, with its corresponding $w^*$, we have*

$$\min_{\tilde{h} \in \tilde{\mathcal{H}}_{n,2}} \sqrt{\sum_{t=1}^{T} \left| h^*(x_{1:t}) - \tilde{h}(x_{1:t}) \right|^2} \leq \frac{4}{\varepsilon_0^2} \sqrt{\frac{\log(\frac{20n}{\delta})}{n}} \, \|w^*\|_2 \, \|x_{1:T}\|_2 \,.$$

- *For any input signal $x_{1:T} \in \ell_\infty$, any $p \geq 1$, and for any dynamical system $h^* \in \mathcal{H}_{\varepsilon_0, M, p}$, with its corresponding $w^*$, we have*

$$\min_{\tilde{h} \in \tilde{\mathcal{H}}_{n,p}} \sqrt[p]{\max_{1 \leq t \leq T} \left| h^*(x_{1:t}) - \tilde{h}(x_{1:t}) \right|^p} \leq \frac{4}{\varepsilon_0^2} \sqrt{\frac{\log(\frac{20n}{\delta})}{n}} \, \|w^*\|_p \, \|x_{1:T}\|_\infty.$$

*Furthermore, if for some $\Lambda > 0$, $h^* \in \mathcal{H}_{\varepsilon_0, M, p, \Lambda}$, the same inequalities hold when the minimization is restricted within $\tilde{\mathcal{H}}_{n,p,\Lambda}$.*

*Proof.* Fix $\delta > 0$. By Lemma 1 with the choice of $d = 2$ and $R = 1 - \varepsilon_0$, we get that with probability at least $1 - \delta$, it holds that

$$\min_i \sup_{|z| \leq 1 - \varepsilon_0} \|Z_i' - z\|_2 \leq \varepsilon' \triangleq 4\sqrt{\frac{\log(\frac{20n}{\delta})}{n}}.$$

So the random numbers $\tilde{\mathcal{Z}} = \{Z_1', \dots Z_n'\}$ induce an $\varepsilon'$-covering of the circle $\{z \in \mathbb{C} : |z| \leq 1 - \varepsilon_0\}$, that is, for any $z$ within this circle, there exists a $\tilde{Z}_{I(z)} \in \tilde{\mathcal{Z}}$ with a distance less than $\varepsilon'$ to $z$. We can evoke Corollary 3 to see that Assumption A1 holds for the choice of $q = r = 2$ and $\varepsilon = \frac{\varepsilon'}{\varepsilon_0^2}$, with a probability at least $1 - \delta$. The same conclusion holds for the choice of $q = r = \infty$.

Proposition 5 shows that when $g(y_{1:t}) = y_t$, for the choice of $p = q = 2$, we can select $L_{2 \to 2} = 1$ in Assumption A2; for the choice of $p = 1$ and $q = 2$, we can select $L_{2 \to 1} = \sqrt{T}$; and for the choice of $p = q = \infty$, we can select $L_{\infty \to \infty} = 1$.

We now use Lemma 4, whose assumptions are satisfied as just shown. For any input signal $x_{1:T} \in \ell_2$ and dynamical system $h^* \in \mathcal{H}_{\varepsilon_0, M, 1}$ (with its corresponding $w^*$), we have

$$\min_{\tilde{h} \in \tilde{\mathcal{H}}_{n,1}} \sum_{t=1}^{T} \left| h^*(x_{1:t}) - \tilde{h}(x_{1:t}) \right| \leq \frac{\varepsilon' \sqrt{T}}{\varepsilon_0^2} \|w^*\|_1 \|x_{1:T}\|_2,$$

with probability at least $1 - \delta$. Likewise, for $h^* \in \mathcal{H}_{\varepsilon_0, M, 2}$, with the same probability we have

$$\min_{\tilde{h} \in \tilde{\mathcal{H}}_{n,2}} \sqrt{\sum_{t=1}^{T} \left| h^*(x_{1:t}) - \tilde{h}(x_{1:t}) \right|^2} \leq \frac{\varepsilon'}{\varepsilon_0^2} \|w^*\|_2 \|x_{1:T}\|_2.$$

For any input signal $x_{1:T} \in \ell_\infty$ and $h^* \in \mathcal{H}_{\varepsilon_0, M, p}$, with the same probability we have

$$\min_{\tilde{h} \in \tilde{\mathcal{H}}_{n,p}} \sqrt[p]{\max_{1 \leq t \leq T} \left| h^*(x_{1:t}) - \tilde{h}(x_{1:t}) \right|^p} \leq \frac{\varepsilon'}{\varepsilon_0^2} \|w^*\|_p \|x_{1:T}\|_\infty.$$

Furthermore, Lemma 4 also implies that when $\|w^*\|_p \leq \Lambda$, the minimizer in $\tilde{\mathcal{H}}_\Lambda$ also satisfies the same inequalities. $\qquad\square$

Note that the choices of (23) and (24) correspond to Case 1 of Section 3. The result for Case 2 is similar, as the non-triviality of the approximation error comes from using randomly selected AR filters, and not from the lagged filters.

## 4.2 Statistical Guarantee

This section develops the statistical guarantee for a time series predictor under general assumptions on the underlying stochastic process. We specifically focus on an ERM-based estimator. We then use this result to provide a guarantee for a time series predictor that uses RPFB to process the time series. Note that Algorithm 1 is not restricted to time series prediction problem, or the use of ERM-based estimator, but the analysis of this section is.

A challenge in analyzing time series data is that the input to the estimator does not satisfy the usual i.i.d. assumption, required by commonly-used tools (e.g., Hoeffding's inequality, etc.) in statistical

learning theory and empirical processes [Györfi et al., 2002; Steinwart and Christmann, 2008]. A common approach to extend those results is by considering mixing processes [Doukhan, 1994], which are stochastic processes that gradually forget their past, and use tools such as the independent block technique [Yu, 1994] to extend the results of i.i.d. processes to mixing processes. Such extensions and analysis have been performed by many prior work.

For example, Meir [2000], used covering number-based analysis for $\beta$-mixing processes and derived guarantees for time series prediction that is adaptive to the model complexity. Mohri and Rostamizadeh [2009] derived Rademacher complexity-based results for $\beta$-mixing processes. Mohri and Rostamizadeh [2010] derived stability-based results for $\beta$ and $\phi$-mixing processes. Farahmand and Szepesvári [2012] provided a finite sample error upper bound for regularized least-squares regression under exponential $\beta$-mixing, with a convergence rate comparable to the rates available for i.i.d. processes, showing that learning under exponential $\beta$-mixing is not significantly slower than learning under the i.i.d. process.

One may summarize all these mixing-based approaches for analyzing learning with dependent stochastic process as follows: We first make certain assumptions on the stochastic process, and then convert it to a similar i.i.d. process with a controlled amount of error. We then use tools developed for i.i.d. processes, including all complexity measures such as covering number, Rademacher complexity, etc., to analyze the derived i.i.d. process.

The shortcoming of this approach is that it decouples the stochastic process and the complexity of the function (hypothesis) space. In this work, we directly use notions of complexity that are tailored to the stochastic process. We follow recent developments in defining various notions of sequential complexities [Rakhlin et al., 2010, 2014] to derive our results. Our results do not require any mixing assumption on the stochastic process. Instead, these assumptions would be incorporated in the definition of the sequential complexity [Rakhlin et al., 2010] and discrepancy [Kuznetsov and Mohri, 2015] of the function space-stochastic process couple.

### 4.2.1 Time Series Prediction Under Function Approximation Error

Consider the time series $(X_1, X_2, \dots)$ with $X_t \in \mathcal{X}$. In this section, $\mathcal{X}$ is a separable metric space. We denote $\mathcal{X}^* = \cup_{t \geq 1} \mathcal{X}^t$. The main object of interest in time series prediction is the conditional expectation of $X_{t+1}$ given $X_{1:t}$, which we denote by $h^*$, i.e.,

$$h^*(X_{1:t}) = \mathbb{E}\left[X_{t+1} | X_{1:t}\right]. \tag{25}$$

We now define the function space to which our estimator belongs. Let $\phi : \mathcal{X}^* \to \mathbb{R}^n$ for some $n \in \mathbb{N}$. This is a set of $n$ filters that accepts any input sequence from $\mathcal{X}$ and returns an $n$-dimensional real-valued vector. A particular example is the RPFB with AR filters defined in Section 3, but in this section we do not specialize to RPFB in this section. For notational simplicity, we may occasionally use $w_t = \phi(x_{1:t})$.

Given a $w \in \mathbb{R}^n$, the function $\psi : \mathbb{R}^n \to \mathcal{H}_0$ defines a feature map from $\mathbb{R}^n$ to a pre-Hilbert space $\mathcal{H}_0$. There is no restriction on the finiteness of the dimension of $\mathcal{H}_0$, so it can be countably infinite. Therefore, we are allowed to think of $\mathcal{H}_0$ (or $\mathcal{H}$ after completion) as an RKHS with a kernel $\kappa(w_1, w_2) = \langle \psi(w_1), \psi(w_2) \rangle_{\mathcal{H}_0}$. In that case, the feature map $\psi$ may be defined only implicitly. We may use $\psi_i$ to refer to the $i$-th component of $\psi$.

Let us define

$$\tilde{h}(x_{1:t}; \alpha) = \tilde{h}(x_{1:t}) = \sum_i \alpha_i \psi_i(\phi(x_{1:t})) = \langle \alpha, \psi(w_t) \rangle.$$

We may also use $\tilde{h}_\alpha$ to refer to $\tilde{h}(\cdot; \alpha)$. Functions $\tilde{h}$ define the following function space

$$\tilde{\mathcal{H}}_\Lambda = \left\{ \tilde{h}_\alpha : \mathcal{X}^* \to \mathbb{R} : \|\alpha\|_2 \leq \Lambda \right\},$$

for a $\Lambda > 0$.

Because some of the tools that we are going to use assume the boundedness of functions involved, we shall focus on a truncated function space instead. Let us define the truncation operator. Consider a real-valued function $f$ defined over a domain $\mathcal{Z}$, which its particular choice shall be specified shortly.

For a fixed $B > 0$, the truncation operator $\mathsf{Tr}_B\left[f\right]$ is defined as

$$\mathsf{Tr}_B\left[f\right](z) \triangleq \begin{cases} f(z) & \text{if } |f(x)| \leq B, \\ \mathrm{sgn}\left(f(z)\right)B & \text{otherwise.} \end{cases}$$

So for $B > 0$ and $\Lambda > 0$, we define

$$\tilde{\mathcal{H}}_{\Lambda,B} = \left\{ \mathsf{Tr}_B\left[\tilde{h}_\alpha\right] : \mathcal{X}^* \to [-B, B] : \|\alpha\|_2 \leq \Lambda \right\}. \tag{26}$$

Consider that we have a sequence $(X_1, X_2, \ldots, X_T, X_{T+1}, X_{T+2})$. By denoting $Y_t = X_{t+1}$, we define $((X_1, Y_1), \ldots, (X_T, Y_T), (X_{T+1}, Y_{T+1}))$. We assume that $|X_t|$ is $B$-bounded almost surely. Define the estimator $\hat{h}$ by solving the following ERM and performing truncation:

$$\hat{h}' \leftarrow \underset{h \in \tilde{H}_\Lambda}{\mathrm{argmin}} \frac{1}{T} \sum_{t=1}^{T} |h(X_{1:t}) - Y_t|^2,$$

$$\hat{h} \leftarrow \mathsf{Tr}_B\left[\hat{h}'\right]. \tag{27}$$

The goal is to provide a guarantee on the closeness of $\hat{h}(X_{1:T+1})$ to $h^*(X_{1:T+1})$.

Here we are focusing on using a squared loss, which is suitable for time series prediction problem, whereas the choice of loss is flexible in Algorithm 1. Also we are particularly focusing on a truncated ERM estimator.

A central object in our result is the notion of discrepancy, introduced by [Kuznetsov and Mohri, 2015]. Discrepancy captures the non-stationarity of the process with respect to the function space.[6]

**Definition 1** (Discrepancy—Kuznetsov and Mohri 2015). *For a stochastic process $X_1, X_2, \ldots$, a function space $\mathcal{H} : \mathcal{X}^* \to \mathbb{R}$, and $T \in \mathbb{N}$, define*

$$\Delta_T(H) \triangleq \sup_{h \in \mathcal{H}} \left\{ \mathbb{E}\left[|h(X_{1:T+1}) - Y_{T+1}|^2 \,|X_{1:T+1}\right] - \frac{1}{T} \sum_{t=1}^{T} \mathbb{E}\left[|h(X_{1:t}) - Y_t|^2 \,|X_{1:t}\right] \right\}.$$

If the value of $T$ is clear from the context, we may use $\Delta(H)$ instead.

The following theorem is the main result of this section.

**Theorem 7.** *Consider the time series $(X_1, \ldots, X_{T+2})$ with $|X_t| \leq B$ almost surely. For $\Lambda > 0$, consider the function space $\tilde{\mathcal{H}}_{\Lambda,B}$ and let the estimator $\hat{h}$ be defined as (27). Assume that $r \triangleq \sup_{w \in \mathbb{R}^n} \|\psi(w)\|_2 < \infty$. Without loss of generality, suppose that $B^2 \Lambda r \geq 1$, $B \geq 1$, and $T \geq 2$. Fix $\delta > 0$. It then holds that there exists a constant $c > 0$ such that with probability at least $1 - \delta$, we have*

$$\left|\hat{h}(X_{1:T+1}) - h^*(X_{1:T+1})\right|^2 \leq \inf_{\tilde{h} \in \tilde{\mathcal{H}}_{\Lambda,B}} \left|\tilde{h}(X_{1:T+1}) - h^*(X_{1:T+1})\right|^2 + cB\Lambda r \log^3(T)\sqrt{\frac{\log(1/\delta)}{T}} + 2\Delta(\tilde{\mathcal{H}}_{\Lambda,B}).$$

*Proof.* Note that for any (measurable) function $h : \mathcal{X}^* \to \mathbb{R}$, we have

$$\mathbb{E}\left[|h(X_{1:t}) - Y_t|^2 \,|X_{1:t}\right] = \mathbb{E}\left[|h(X_{1:t}) - h^*(X_{1:t})|^2 \,|X_{1:t}\right] + \mathbb{E}\left[|h^*(X_{1:t}) - Y_t|^2 \,|X_{1:t}\right] + 2\mathbb{E}\left[(h(X_{1:t}) - h^*(X_{1:t}))\left(h^*(X_{1:t}) - Y_t\right) |X_{1:t}\right]$$

$$= |h(X_{1:t}) - h^*(X_{1:t})|^2 + \mathbb{E}\left[|h^*(X_{1:t}) - Y_t|^2 \,|X_{1:t}\right]$$

where we used the fact that $h(X_{1:t}) - h^*(X_{1:t})$ is $\sigma(X_{1:t})$-measurable and that by the definition of $h^*$, the inner product term is zero, because

$$\mathbb{E}\left[(h(X_{1:t}) - h^*(X_{1:t}))\left(h^*(X_{1:t}) - Y_t\right) |X_{1:t}\right] = (h(X_{1:t}) - h^*(X_{1:t}))\left(h^*(X_{1:t}) - \mathbb{E}\left[Y_t|X_{1:t}\right]\right)$$
$$= 0.$$

So we have the following function (or filter) approximation and estimation error decomposition:

$$\left|\hat{h}(X_{1:T+1}) - h^*(X_{1:T+1})\right|^2 =$$

$$\left(\mathbb{E}\left[\left|\hat{h}(X_{1:T+1}) - Y_{T+1}\right|^2 \Big| X_{1:T+1}\right] - \inf_{\tilde{h}\in\tilde{\mathcal{H}}_{\Lambda,B}} \mathbb{E}\left[\left|\tilde{h}(X_{1:T+1}) - Y_{T+1}\right|^2 \Big| X_{1:T+1}\right]\right) +$$

$$\left(\inf_{\tilde{h}\in\tilde{\mathcal{H}}_{\Lambda,B}} \mathbb{E}\left[\left|\tilde{h}(X_{1:T+1}) - Y_{T+1}\right|^2 \Big| X_{1:T+1}\right] - \mathbb{E}\left[\left|h^*(X_{1:T+1}) - Y_{T+1}\right|^2 \Big| X_{1:T+1}\right]\right)$$

$$= \mathrm{est}(T, \tilde{\mathcal{H}}_{\Lambda,B}) + \inf_{\tilde{h}\in\tilde{\mathcal{H}}_{\Lambda,B}}\left|\tilde{h}(X_{1:T+1}) - h^*(X_{1:T+1})\right|^2. \tag{28}$$

The first term is the estimation error, and the second is the approximation error of the function space $\tilde{\mathcal{H}}_{\Lambda,B}$. We now provide an upper bound on the estimation error.

$$\mathrm{est}(T, \tilde{\mathcal{H}}_{\Lambda,B}) = \sup_{\tilde{h}\in\tilde{\mathcal{H}}_{\Lambda,B}} \left\{\mathbb{E}\left[\left|\hat{h}(X_{1:T+1}) - Y_{T+1}\right|^2 \Big| X_{1:T+1}\right] - \mathbb{E}\left[\left|\tilde{h}(X_{1:T+1}) - Y_{T+1}\right|^2 \Big| X_{1:T+1}\right]\right\} = \tag{29}$$

$$\sup_{\tilde{h}\in\tilde{\mathcal{H}}_{\Lambda,B}} \left\{\mathbb{E}\left[\left|\hat{h}(X_{1:T+1}) - Y_{T+1}\right|^2 \Big| X_{1:T+1}\right] - \frac{1}{T}\sum_{t=1}^{T}\mathbb{E}\left[\left|\hat{h}(X_{1:t}) - Y_t\right|^2 \Big| X_{1:t}\right] \right. \tag{30}$$

$$+ \frac{1}{T}\sum_{t=1}^{T}\mathbb{E}\left[\left|\hat{h}(X_{1:t}) - Y_t\right|^2 \Big| X_{1:t}\right] - \frac{1}{T}\sum_{t=1}^{T}\left|\hat{h}(X_{1:t}) - Y_t\right|^2 \tag{31}$$

$$+ \frac{1}{T}\sum_{t=1}^{T}\left|\hat{h}(X_{1:t}) - Y_t\right|^2 - \frac{1}{T}\sum_{t=1}^{T}\left|\tilde{h}(X_{1:t}) - Y_t\right|^2 \tag{32}$$

$$+ \frac{1}{T}\sum_{t=1}^{T}\left|\tilde{h}(X_{1:t}) - Y_t\right|^2 - \frac{1}{T}\sum_{t=1}^{T}\mathbb{E}\left[\left|\tilde{h}(X_{1:t}) - Y_t\right|^2 \Big| X_{1:t}\right] \tag{33}$$

$$\left. + \frac{1}{T}\sum_{t=1}^{T}\mathbb{E}\left[\left|\tilde{h}(X_{1:t}) - Y_t\right|^2 \Big| X_{1:t}\right] - \mathbb{E}\left[\left|\tilde{h}(X_{1:T+1}) - Y_{T+1}\right|^2 \Big| X_{1:T+1}\right]\right\}. \tag{34}$$

The term (32), $\frac{1}{T}\sum_{t=1}^{T}|\hat{h}(X_{1:t}) - Y_t|^2 - \frac{1}{T}\sum_{t=1}^{T}|\tilde{h}(X_{1:t}) - Y_t|^2$, is less than or equal to zero. To see this, we write it as

$$\frac{1}{T}\sum_{t=1}^{T}\left|\hat{h}(X_{1:t}) - Y_t\right|^2 - \frac{1}{T}\sum_{t=1}^{T}\left|\tilde{h}(X_{1:t}) - Y_t\right|^2 =$$

$$\left(\frac{1}{T}\sum_{t=1}^{T}\left|\hat{h}(X_{1:t}) - Y_t\right|^2 - \frac{1}{T}\sum_{t=1}^{T}\left|\hat{h}'(X_{1:t}) - Y_t\right|^2\right) +$$

$$\left(\frac{1}{T}\sum_{t=1}^{T}\left|\hat{h}'(X_{1:t}) - Y_t\right|^2 - \frac{1}{T}\sum_{t=1}^{T}\left|\tilde{h}(X_{1:t}) - Y_t\right|^2\right).$$

Recall that $\hat{h} = \mathsf{Tr}_B\left[\hat{h}'\right]$. Since $|Y_t| \le B$ a.s., $\left|\mathsf{Tr}_B\left[\hat{h}'(X_{1:t})\right] - Y_t\right| \le |\hat{h}'(X_{1:t}) - Y_t|$, so the first term on the RHS is non-positive. For the second term, recall that $\hat{h}'$ is the minimizer of $\frac{1}{T}\sum_{t=1}^{T}|h(X_{1:t}) - Y_t|^2$ within $\tilde{H}_\Lambda$. So the empirical loss with $\tilde{h}'$ is no larger than the value of the loss with any other function $h \in \tilde{H}_\Lambda$, including those in $\tilde{\mathcal{H}}_{\Lambda,B} \subseteq \tilde{\mathcal{H}}_\Lambda$.

Because both $\tilde{h}$ and $\hat{h}$ are in $\tilde{\mathcal{H}}_{\Lambda,B}$, we can upper bound terms (30) and (34), as well as terms (31) and (33) with their supremum in $\tilde{\mathcal{H}}_{\Lambda,B}$ to get that

$$
\mathrm{est}(T, \tilde{\mathcal{H}}_{\Lambda,B}) \leq 2 \sup_{h \in \tilde{\mathcal{H}}_{\Lambda,B}} \left\{ \mathbb{E}\left[ |h(X_{1:T+1}) - Y_{T+1}|^2 \,|\, X_{1:T+1} \right] - \frac{1}{T} \sum_{t=1}^{T} \mathbb{E}\left[ |h(X_{1:t}) - Y_t|^2 \,|\, X_{1:t} \right] \right\} +
$$
$$
2 \sup_{h \in \tilde{\mathcal{H}}_{\Lambda,B}} \left\{ \frac{1}{T} \sum_{t=1}^{T} \left( \mathbb{E}\left[ |h(X_{1:t}) - Y_t|^2 \,|\, X_{1:t} \right] - |h(X_{1:t}) - Y_t|^2 \right) \right\}. \tag{35}
$$

The first term is the discrepancy $\Delta(\tilde{\mathcal{H}}_{\Lambda,B})$. The second term is the supremum of a martingale equivalent of an empirical process, which we shall upper bound next.

Define the following function space:

$$
\mathcal{G}_{\Lambda,B} \triangleq \left\{ g(x_{1:t}, y_{1:t}) = |\mathsf{Tr}_B\left[ \langle\, \alpha \,,\, \psi(\phi(x_{1:t})) \,\rangle \right] - y_t|^2 \,:\, \|\alpha\|_2 \leq \Lambda \right\}. \tag{36}
$$

By Lemma 10 in Appendix A.2, upon satisfaction of its conditions, we have that

$$
\mathbb{P}\left\{ \sup_{h \in \tilde{\mathcal{H}}_{\Lambda,B}} \left| \frac{1}{T} \sum_{t=1}^{T} \left( \mathbb{E}\left[ |h(X_{1:t}) - Y_t|^2 \,|\, X_{1:t} \right] - |h(X_{1:t}) - Y_t|^2 \right) \right| > \alpha \right\} \leq
$$
$$
8L \exp\left( -\frac{\alpha^2}{c \ln^3(T) \mathbf{Rad}_T^2(\mathcal{G}_{\Lambda,B})} \right), \tag{37}
$$

for some absolute constant $c > 0$ and with

$$
L = \max\left\{ e^4, \sum_{j \geq 1} [\mathcal{N}_\infty(2^{-j}, \mathcal{G}_{\Lambda,B}, T)]^{-1} \right\}.
$$

To choose $L$, we evoke Proposition 12 in Appendix A.3, which indicates that $\mathcal{N}_\infty(\varepsilon, \mathcal{G}_{\Lambda,B}, T) \geq \frac{2B^2\Lambda r}{\varepsilon}$. So $\sum_{j \geq 1} [\mathcal{N}_\infty(2^{-j}, \mathcal{G}_{\Lambda,B}, T)]^{-1} \leq 1$ whenever $B^2\Lambda r \geq 1$. So we may choose $L = e^4$.

To verify the conditions of Lemma 10, note that by the same proposition, the condition $\mathcal{N}_\infty(2^{-1}, \mathcal{G}_{\Lambda,B}, T) \geq 4$ is satisfied. Moreover, because $g_0(x_{1:t}, y_{1:t}) = |\mathsf{Tr}_B\left[ \langle\, 0 \,,\, \psi(\phi(x_{1:t})) \,\rangle \right] - y_t|^2 = y_t^2$ belongs to $\mathcal{G}_{\Lambda,B}$, we may use the Khintchine inequality to conclude that

$$
\mathbf{Rad}_T(\mathcal{G}_{\Lambda,B}) \geq \sup_{(x_{1:T}, y_{1:T})} \mathbb{E}_{\varepsilon_{1:T}}\left[ \frac{1}{T} \sum_{t=1}^{T} \varepsilon_t g_0(x_{1:t}(\varepsilon_{1:T}), y_t(\varepsilon_{1:T})) \right]
$$
$$
\geq \frac{1}{T} \sup_{y_{1:T}} A_1 \sqrt{\sum_{t=1}^{T} |y_t|^4} = \frac{A_1 B^2}{\sqrt{T}},
$$

in which $A_1$ is the constant of the lower bound of the Khintchine inequality and can be set to $\frac{1}{\sqrt{2}}$ [Haagerup, 1981]. So the condition $\mathbf{Rad}_T(\mathcal{G}_{\Lambda,B}) \geq \frac{1}{T}$ is also satisfied when $T \geq 2$ and $B \geq 1$.

To upper bound $\mathbf{Rad}_T(\mathcal{G}_{\Lambda,B})$ in the exponential bound (37), we use Proposition 11 in Appendix A.3, which shows that

$$
\mathbf{Rad}_T(\mathcal{G}_{\Lambda,B}) \leq 32B \left( 1 + 4\sqrt{2} \ln^{3/2}(eT^2) \right) \frac{\Lambda r}{\sqrt{T}}.
$$

Now pick a fixed $\delta > 0$, and let the probability of the failure event in the LHS of (37) be equal to $\delta$. Solve for $\delta$ to obtain that with probability more than $1 - \delta$, it holds that

$$
\sup_{h \in \hat{\mathcal{H}}_{\Lambda,B}} \left| \frac{1}{T} \sum_{t=1}^{T} \left( \mathbb{E}\left[ |h(X_{1:t}) - Y_t|^2 \,|\, X_{1:t} \right] - |h(X_{1:t}) - Y_t|^2 \right) \right|
$$

$$
\leq \mathbf{Rad}_T(\mathcal{G}_{\Lambda,B}) \ln^{3/2}(T) \sqrt{c \ln\left( \frac{8e^4}{\delta} \right)}
$$

$$
\leq 32B(1 + 4\sqrt{2} \ln^{3/2}(T)) \ln^{3/2}(T) \sqrt{c \ln\left( \frac{8e^4}{\delta} \right)} \frac{\Lambda r}{\sqrt{T}}
$$

$$
\leq c_1 B \Lambda r \log^3(T) \sqrt{\frac{\log\left( \frac{1}{\delta} \right)}{T}}, \tag{38}
$$

for some absolute constant $c_1 > 0$.

The error decomposition (28) alongside upper bounds (35) and (38) lead to the desired result. $\qquad\square$

### 4.3 Finite Sample Error Bound for RPFB-based Time Series Predictor

After developing the filter approximation result of Section 4.1.4 and the statistical guarantee of Section 4.2.1, we are ready to provide a finite-sample error upper bound for a RPFB-based estimator. Here we consider univariate time series, i.e., $\mathcal{X} = \mathbb{R}$.

We consider the case that the conditional expectation of $X_{t+1}$ given $X_{1:t}$, the function $h^*(X_{1:t})$ (25), belongs to $\mathcal{H}_{\varepsilon_0, M, 2, \Lambda}$ (23).

Recall that for RPFB, $\phi : \mathcal{X}^* \to \mathbb{R}^n$ is

$$
\phi(x_{1:t}) = \left[ \left( \frac{1}{1 - Z_1' z^{-1}}, \ldots, \frac{1}{1 - Z_n' z^{-1}} \right) * x_{1:t} \right]_t,
$$

where notation $[\cdot]_t$ means that we take the last value of the output sequence. The next theorem is the main theoretical result of this paper.

**Theorem 8.** *Consider the time series $(X_1, \ldots, X_{T+2})$, and assume that $|X_t| \leq B$ (a.s.). Without loss of generality suppose that $B \geq 1$. Let $0 < \varepsilon_0 < 1$, $M \in \mathbb{N}$, and $\Lambda > 0$ and assume that the conditional expectation $h^*(X_{1:t}) = \mathbb{E}\left[ X_{t+1} | X_{1:t} \right]$ belongs to the class of linear filters $\mathcal{H}_{\varepsilon_0, M, 2, \Lambda}$ (23). Set an integer $n \geq M$ for the number of random projection filters and let $\tilde{\mathcal{H}}_\Lambda = \tilde{\mathcal{H}}_{n, 2, \Lambda}$ (24) and the truncated space be $\tilde{\mathcal{H}}_{\Lambda, B}$ (26). Consider the estimator $\hat{h}$ that is defined as (27). Without loss of generality assume that $\Lambda \geq \frac{\varepsilon_0}{B^2 \sqrt{n}}$ and $T \geq 2$. Fix $\delta > 0$. It then holds that there exists constants $c_1, c_2 > 0$ such that with probability at least $1 - \delta$, we have*

$$
\left| \hat{h}(X_{1:T+1}) - h^*(X_{1:T+1}) \right|^2 \leq \frac{c_1 B^2 \Lambda}{\varepsilon_0} \log^3(T) \sqrt{\frac{n \log(1/\delta)}{T}} + \frac{c_2 B^2 \Lambda^2}{\varepsilon_0^4} \frac{\log\left( \frac{20n}{\delta} \right)}{n} + 2\Delta(\tilde{\mathcal{H}}_{\Lambda, B}).
$$

*Proof.* We start by obtaining an upper bound on $r = \sup_{w \in \mathbb{R}^n} \|\psi(w)\|_2 < \infty$ in Theorem 7. The mapping $\psi$ (Section 4.2.1) is

$$
\psi(\phi(x_{1:t})) = (\phi_1(x_{1:t}), \ldots, \phi_n(x_{1:t})),
$$

so we first provide an upper bound on $\phi_i(x_{1:t})$ for a $B$-bounded sequence $x_{1:t}$. Consider the output of a filter $\phi(x_{1:t}) = [h_z * x_{1:t}]_t$, corresponding to a filter $h_z$ specified by a pole at $z = \rho e^{j\theta}$ with $0 \leq \rho < 1$. By Young's inequality for convolutions, we have

$$
|\phi(x_{1:t})| \leq \|h_z * x_{1:t}\|_\infty \leq \|h_z\|_1 \|x_{1:t}\|_\infty \leq B \sum_{t \geq 0} |\rho e^{j\theta}|^t = \frac{B}{1 - \rho}. \tag{39}
$$

By construction $\rho < 1 - \varepsilon_0$, so we can set

$$
r = \frac{B\sqrt{n}}{\varepsilon_0}. \tag{40}
$$

As $|X_t| \leq B$, the function $h^*(X_{1:t}) = \mathbb{E}\left[X_{t+1}|X_{1:t}\right]$ is also $B$-bounded. Therefore, by the truncation argument, for any $h$, we have

$$\left|\mathsf{Tr}_B\left[h(X_{1:T+1})\right] - h^*(X_{1:T+1})\right|^2 \leq \left|h(X_{1:T+1}) - h^*(X_{1:T+1})\right|^2.$$

Let $h'$ be the minimizer of the approximation loss within $\tilde{\mathcal{H}}_\Lambda$, i.e.,

$$h' \leftarrow \inf_{\tilde{h} \in \tilde{\mathcal{H}}_\Lambda} \left|\tilde{h}(X_{1:T+1}) - h^*(X_{1:T+1})\right|^2.$$

The function $h'$ is not necessarily $B$-bounded, so we consider its truncation at the magnitude of $B$. Notice that the truncated function $\mathsf{Tr}_B\left[h'\right]$ is in $\tilde{\mathcal{H}}_{\Lambda,B}$. The truncated function, however, is not necessarily the minimizer of the approximation loss within $\tilde{\mathcal{H}}_\Lambda$, but it provides an upper bound to the loss. This argument and Theorem 6 indicate that for any $\delta_1 > 0$, and for any $h^* \in \mathcal{H}_{\varepsilon_0,M,2,\Lambda}$ (with its corresponding $w^*$), we have

$$\inf_{\tilde{h} \in \tilde{\mathcal{H}}_{\Lambda,B}} \left|\tilde{h}(X_{1:T+1}) - h^*(X_{1:T+1})\right|^2 \leq \left|\mathsf{Tr}_B\left[h'\right](X_{1:T+1}) - h^*(X_{1:T+1})\right|^2$$

$$\leq \left|h'(X_{1:T+1}) - h^*(X_{1:T+1})\right|^2$$

$$= \inf_{\tilde{h} \in \tilde{\mathcal{H}}_\Lambda} \left|\tilde{h}(X_{1:T+1}) - h^*(X_{1:T+1})\right|^2$$

$$\leq \min_{\tilde{h} \in \tilde{\mathcal{H}}_\Lambda} \max_{1 \leq t \leq T+1} \left|h^*(x_{1:t}) - \tilde{h}(x_{1:t})\right|^2$$

$$\leq \frac{16\log\left(\frac{20n}{\delta_1}\right)}{\varepsilon_0^4 n} \|w^*\|_2^2 \|x_{1:T+1}\|_\infty^2$$

$$\leq \frac{16\log\left(\frac{20n}{\delta_1}\right)}{\varepsilon_0^4 n} \Lambda^2 B^2, \tag{41}$$

with probability at least $1 - \delta_1$.

We apply Theorem 7, with the choice of $r$ (40), and with the upper bound (41) on the filter approximation error. We obtain that for any $\delta_1, \delta_2 > 0$, there exists a constant $c > 0$ such that

$$\left|\hat{h}(X_{1:T+1}) - h^*(X_{1:T+1})\right|^2 \leq \inf_{\tilde{h} \in \tilde{\mathcal{H}}_{\Lambda,B}} \left|\tilde{h}(X_{1:T+1}) - h^*(X_{1:T+1})\right|^2 + cB\Lambda r \ln^3(T)\sqrt{\frac{\ln(1/\delta_2)}{T}} +$$

$$2\Delta(\tilde{\mathcal{H}}_{\Lambda,B})$$

$$\leq \frac{16B^2\Lambda^2}{\varepsilon_0^4}\frac{\log\left(\frac{20n}{\delta_1}\right)}{n} + c\frac{B^2\Lambda}{\varepsilon_0}\ln^3(T)\sqrt{\frac{n\ln(1/\delta_2)}{T}} + 2\Delta(\tilde{\mathcal{H}}_{\Lambda,B}).$$

with probability at least $1 - (\delta_1 + \delta_2)$. Setting $\delta_1 = \delta_2 = \delta/2$ finishes the proof. $\qquad\square$

The upper bounds has three terms: estimation error, filter approximation error, and the discrepancy. The term $\frac{B^2\Lambda}{\varepsilon_0}\log^3(T)\sqrt{\frac{n\log(1/\delta)}{T}}$ corresponds to the estimation error. It decreases as the length $T$ of the time series increases. As we increase the number of filters $n$, the upper bounds shows an increase of the estimation error. This is a manifestation of the effect of the input dimension on the error of the estimator.

The second term $\frac{B^2\Lambda^2}{\varepsilon_0^4}\frac{\log\left(\frac{20n}{\delta}\right)}{n}$ provides an upper bound to the filter approximation error. It shows that the error decreases as we add more filters. This indicates that RPFB provides a good approximation to the space of the dynamical systems $\mathcal{H}_{\varepsilon_0,M,2,\Lambda}$ (23).

Both terms have a proportional dependence on the magnitude $B$ of the random variables in the time series. Also as the minimum distance $\varepsilon_0$ of the poles to the unit circle decreases, the upper bound increases too. This can be intuitively understood by recalling that as the magnitude of a pole of a filter

gets closer to 1, the filter decays slower, and it behaves similar to an integrator. In that case, given proper input signal, the size of the feature can become large, as can be seen in (39). Another related reason is that poles becomes more sensitive to perturbation, and thus approximation by another pole, as we get close to the unit circle, as shown by Lemma 2 and Corollary 3.

Finally, the discrepancy term $\Delta(\tilde{\mathcal{H}}_{\Lambda,B})$ captures the non-stationary of the process, and has been discussed in detail by Kuznetsov and Mohri [2015]. Understanding the conditions when the discrepancy gets close to zero is an interesting topic for future research.

By setting the number of RP filters to $n = \frac{T^{1/3}\Lambda^{2/3}}{\varepsilon_0^2}$, and under the condition that $\Lambda \leq T$, we can simplify the upper bound to

$$\left|\hat{h}(X_{1:T+1}) - h^*(X_{1:T+1})\right|^2 \leq \frac{cB^2\Lambda^{4/3}\log^3(T)\sqrt{\log(\frac{1}{\delta})}}{\varepsilon_0^2 T^{1/3}} + 2\Delta(\tilde{\mathcal{H}}_{\Lambda,B}),$$

which holds with probability at least $1 - \delta$, for some constant $c > 0$. As $T \to \infty$, the error converges to the level of discrepancy term.

*Remark* 5. Theorem 8 is stated for the choice of linear dynamical system space $\mathcal{H}_{\varepsilon_0,M,2,\Lambda}$ (23), which corresponds to Case 1 in Section 3. A similar result holds for Case 2 as well. The reason is that the proof of this theorem uses Theorem 7, which does not depend on the particular choice of space $\tilde{\mathcal{H}}_{\Lambda,B}$, so one can incorporate additional time-lagged features of (11). As long as the number of time-lagged features matches (or exceeds) the true filter's, we do not have any additional filter approximation error.

*Remark* 6. The values of $M$, $\varepsilon_0$, and $\Lambda$ of the true dynamical system space $\mathcal{H}_{\varepsilon_0,M,2,\Lambda}$ are often unknown, so the choice of number of filters $n$ in RPFB, the size of the space $M$, etc. cannot be selected based on them. Instead one should use a model selection procedure to pick the appropriate values of the parameters, either by using a validation set or by a complexity regularization procedure. The complexity regularization for dependent data is explored by Meir [2000], albeit under known mixing condition.

# 5    Experiments

In this section we empirically study RPFB on a range of time-series-related problem. First we focus on the time series prediction problem of an ARMA process (Section 5.1). Afterwards, we study the fault detection problem of ball bearings (Section 5.2). This is an example of an industrial problem where the computational resources might be limited. We compare RPFB with the fixed-window history-based approach, and also report some preliminary results on the application of LSTM for that problem. Our results show the competitiveness of RPFB. Finally, we compare RPFB and the fixed-window history-based approach for the task of heart rate classification problem (Section 5.2).

## 5.1    Time Series Prediction of an ARMA Process

In order to show the effectiveness of the RPFB approach, we start with predicting the next time-step value of an ARMA time series. Our goal is to find a function $\hat{f}$ that predicts $X_{t+1}$ given the values of the time series in previous steps, i.e., $X_{1:t}$.

The time series generating process is an ARMA process, characterized by the location of its zero $-z$:

$$X_t = \frac{(1 + \mathsf{z}z^{-1})}{(1 - 0.6z^{-1})}U_t, \ U_t \sim N(0,1). \tag{42}$$

We apply two approaches for feature extraction: fixed-window history-based approach and RPFB. In the former approach, we use a sliding window with length $H$, that is, we use the feature vector $Z_i = X_{i-H+1:i}$ for $i = H, \ldots, T$.

For the RPFB, we first randomly draw $n$ stable degree 1 (real pole) or degree 2 (a complex conjugate pair of poles) filters in order to create a filter bank with stable auto-regressive filters. More specifically, a fraction of the filters are selected to be of degree 1, for which the location of the $j$-th pole $Z'_j = r$ is selected by drawing $r$ from a uniform distribution over $[-1, +1]$. For degree 2 filters, which are

Figure 1: (ARMA time series) The prediction error using RPFB and fixed-window features for different location of the (moving average) zero as a function of the number of features.

described by the conjugate pairs $Z'_j = re^{j\theta}$ and $\bar{Z}'_j = re^{-j\theta}$, we choose $r$ from a uniform distribution over $[0, 1]$, and $\theta$ from a uniform distribution over $[0, 2\pi]$ (cf. Remark 1).[7] The time series $X_{1:T}$, generated from (42) in a manner to be described shortly, is then passed through all such filters to construct the feature set of

$$Z_i = \left( \tilde{X}_{1,i}, \cdots \tilde{X}_{n,i} \right) \qquad i = 1, \ldots, T.$$

Here $\tilde{X}_{k,i}$ denotes the filtered signal at time $i$ obtained by the convolution of the time series and the impulse response of the $k$-th filter, i.e. $\tilde{X}_{k,1:T} = h_k(t) * X_{1:T}$.

We then use a standard regression algorithm over both feature sets to perform the next step prediction task and compare their performance. In this example we use ridge regression as the estimator.

For each fixed value of z, we generate 20 independent time series with the length of $T = 10000$ from the process (42). For each time series (or signal), we generate multiple RPFBs containing different numbers of filters from the same random seed. Thus, a RPFB with fewer filters is a subset of a RPFB with more filters for each signal. We use different random seeds for generating RPFBs for different signals. We then pass each ARMA time series through all its corresponding RPFBs with different filter sizes. Creating both RPFB and fixed-window feature sets, we then divide each feature set into training and testing datasets by assigning the first 6000 data points of a time series to the training dataset and the rest 4000 of data points to the testing dataset. We investigate the predication error, in the $\ell_2$-norm, for different zero locations by changing the value of z in (42).

Figure 1 depicts the mean of the prediction error, for both RPFB and fixed-window history-based approaches, as the number of features (filters for RPFB; size of history for the window-based approach) varies and the location of zero z is changing. We observe that as the number of features increases, the prediction error decreases to 1, the optimal value, for both feature extraction methods. This indicates that both methods can provide a reasonably good summary of the time series for this time series prediction task.

We also observe that compared to the fixed-window features, the ridge regression with RPFB performs better when the ARMA's zero (i.e., the root of the moving average term) is close to the unit circle (z = 0.99, z = 0.95) and it reaches close to the optimal error with fewer number of features. On the other hand, when the ARMA's zero moves farther from the unit circle (z = 0.9, z = 0.8), the performance of ridge regression on both feature sets becomes comparable, i.e., with almost the same number of features (e.g., $n = H = 11$), they both have very small errors. This can be understood

Figure 2: (ARMA time series) The prediction error using RPFB and fixed-window features for different number of features as a function of the number of training points (signal length). The dashed line is the optimal error obtained using optimal estimator (6). The error bars shows one standard error.

better by noticing that the zero of the process would become the pole of the optimal estimator, so whenever the magnitude of the zero becomes close to one, the response of the optimal estimator, which depends on the location of the pole, decays slower. Therefore, by ignoring the values of the distant past, the fixed-window history-based approach would lose more information about the process. See the discussion in the paragraph just after (3) and the Section 2 in general.

### 5.1.1 Effect of Number of Samples

In this section we study the effect of number of samples in the training set on the prediction error. Using the same ARMA process as in Section 5.1 with z = 0.99, we generate multiple time series with different lengths. For each length, we generate 64 independent time series. To generate RPFB feature set we pass signals with the same length through different RPFBs generated using different random seeds. Figure 2 shows the empirical mean of the prediction error vs. the size of the training set for different feature sizes of both RPFB and fixed-window history-based method. As before, we use ridge regression as the estimator. The error bars on the curve show one standard error around the mean. The error obtained using the optimal filter (6) is also depicted as a dashed horizontal line.

As expected, the prediction error decreases with the increase in the size of the training set as well as the number of features. Another observation is that for small number of filters in RPFB ($n = 3$) the error bar is large, i.e., RPFB has a high variance. This is because it is less likely to capture the dynamic of the system with only a few filters. But as we increase the number of features, RPFB's performance becomes comparable or even slightly better than history-based approach.

### 5.2 Fault Detection: Condition Monitoring for Bearings

Reliable operation of rotating equipments (e.g., turbines) depends on the condition of their bearings, which makes the detection of whether a bearing is faulty and requires maintenance of crucial importance. We consider a bearing vibration dataset provided by Machinery Failure Prevention Technology (MFPT) Society in our experiments.[8] Fault detection of bearings is an example of industrial applications where the computational resources are limited, and fast methods are required, e.g., only a micro-controller or a cheap CPU, and not a GPU, might be available.

Figure 3: (Bearing Dataset) Classification error on the test dataset using RPFB and fixed-window history-based feature sets. The RPFB results are averaged over 20 independent randomly selected RPFB. The error bars show one standard error.

The dataset consists of three univariate time series corresponding to a baseline (good condition/class 0), an outer race fault (class 1), and inner race fault (class 2). The goal is to find a classifier that predicts the class label at the current time $t$ given the vibration time series $X_{1:t}$. In a real-world scenario, we train the classifier on a set of previously recorded time series, and later let it operate on a new time series observed from a device. The goal would be to predict the class label at each time step as new data arrives. Here, however, we split each of three time series to a training and testing subsets. More concretely, we first pass each time series through RPFB (or define a fixed-window of the past $H$ values of them). We then split the processed time series, which has the dimension of the number of RPFB or the size of the window, to the training and testing sets. We select the first 3333 time steps to define the training set, and the next 3333 data points as the testing dataset. As we have three classes, this makes the size of training and testing sets both equal to 10K. We process each dimension of the features to have a zero mean and a unit variance for both feature types. We perform 20 independent runs of RPFB, each of which with a new set of randomly selected filters.

Figure 3 shows the classification error of three different classifier (Logistic Regression (LR) with the $\ell_2$ regularization, Random Forest (RF), and Support Vector Machine (SVM) with Gaussian kernel) on both feature types, with varying feature sizes. We observe that as the number of features increases, the error of all classifiers decreases too. It is also noticeable that the error heavily depends on the type of classifier, with SVM being the best in the whole range of number of features. The use of RPFB instead of fixed-window history-based one generally improves the performance of LR and SVM, but not for RF. Refer to Appendix B for more detail on the experiment.

The filters in RPFB are not adapted to data, but are randomly selected independent of the data. Therefore, it is possible that a method that adjusts filters data-dependently extract more suitable features from the time series. To test how much better such a method might be, and see whether RPFB is still a reasonable feature extractor or not when the computational cost is not the main consideration, we provide some preliminary results of applying LSTM to the very same problem of bearings fault detection.

Figure 4 shows the classification error as a function of the number of filters (i.e., LSTM units). The difference between three columns is the size of the input sequence window given to the LSTM module before it has to make a prediction. This is similar to the fixed-window history-based approach with the difference that we allow the feature extractor to process the window sequentially and construct

Figure 4: (Bearing Dataset) Classification error on the test dataset using LSTM with varying number of filters and different sizes of the input sequence. The results are averaged over 20 independent runs. The error bars show one standard error.

an internal state to summarize the sequence. The length of the input sequence window is selected to be either 100, 200, or 400. There are two curves in each figure. The solid one corresponds to an architecture with several LSTM units (corresponding to Filters No in the graphs) whose outputs are connected to the 3 output of the network using a fully-connected (FC) layer with softmax activation. The loss is log loss, i.e., cross-entropy. The dashed one corresponds to a slightly deeper architecture: the output of the LSTM units are connected to a FC layer with 100 units (with ReLu activations), which is then connected to 3 outputs. For each setting of the parameters and architectures, we run the experiment 20 times to measure the variability of the results, and we report the average. The error bars show one standard error across the average. Appendix B.4 describes some detail of the optimization procedure.

There are several notable observations. The first is that the performance of LSTM at each run is highly variable, which is due to the stochastic nature of the optimizer and the possibility of getting to different regions of the parameters space because of the non-convexity of the loss function w.r.t. the parameters of the network. This variability is especially noticeable for larger input sequence sizes (200 and 400). Moreover, the performance of the deeper architecture seem to be generally better than the shallower one. Finally, the LSTM-based estimators do not outperform SVM or RF. In fact, SVM with RPFB achieves the error less than 0.1, whereas this does not happen by any of the LSTM-based estimators. One should notice that we have not extensively optimized the hyper-parameters of the LSTM-based networks; so it is possible that with better choices of parameters, we could obtain better performance.

One of the premises of RPFB is that it is computationally fast: the computation time per sample is linear in the number of features. To understand the relative cost of RPFB compared to other parts of the estimation, we record the computation time of all the methods used within the bearing experiments. Figure 5 shows the result. The reported time is the average time per run, and consists of all steps of the pipeline, except the loading of the data: extracting features (for RPFB or fixed-window history-based approach), preprocessing, initializing the model (which is relatively noticeable for LSTM), model selection (for LR and SVM only), training the model, and evaluating the test error. RF is clearly the fastest, partly because we do not perform any model selection. SVM is slow, as we do an extensive model selection (225 different models for each choice of features). LSTM is especially slow, particularly noting that we do not perform any hyper-parameter search, and it is running on a GPU.

Since the computation time of RPFB is independent of the estimator, it should be less than that of RF (RPFB). So at least we can claim that for a method such as LR or SVM, the computational cost of passing data through RPFB is not the dominant factor. To get a more accurate estimate, we pass a larger fraction of the same dataset with 300K number of samples through RPFB with varying number of filters (compare this with 20K samples of the previous experiments). Here we do not fit

Figure 5: (Bearing Dataset) Computation time of various estimation methods as a function of the number of features.

Figure 6: (Bearing Dataset) Computation time of RPFB for a time series with $300K$ data points. The left figure shows the computation time as a function of the number of filters. The right figure shows the number of samples-filter processed per second.

the estimators anymore. The results are depicted in the left hand side of Figure 6. As expected, the computational cost increases almost linearly as the number of filters increases. For example, we observe that RPFB can process 300K data points and generate 1000 filters in about 10 seconds. On the right hand side, we report the number of samples-filter that can be processed per second. The number is approximately 25M samples-filter/second, which is relatively fast, e.g., one can generate a thousand features at the rate of 25K samples per second. Refer to Appendices B.1 and B.2 for the detail of software and hardware used for this part of the experiment.

Considering these performance comparisons between RPFB and other solutions, including LSTM, and noticing that RPFB as a feature extraction method is fast (and can easily be parallelized or implemented on cheap specially-designed hardware, which is not the case for LSTM), one might suggest that RPFB is a viable approach for feature extraction from time series.

Figure 7: (Heart Rate Dataset) Classification error on the test dataset using RPFB feature sets and fixed-window feature sets.

## 5.3 Condition Monitoring for Heart Rate

To show the flexibility of the RPFB approach to analyze time series data, we conduct experiments on a problem from a completely different domain: heart rate classification. This dataset is provided by MIT-BIH database distribution.[9] It includes 4 time series corresponding to 4 subjects. Each time series records evenly-spaced measurements of instantaneous heart rate from each subject. While subjects were involving in similar activities, the measurements occur at 0.5 second intervals. We apply the same methodology as we did for the bearing dataset. Figure 7 shows the error rates of different classifiers (LR, SVM, RF) with either the RPFB or fixed-window feature sets.

The error rates of SVM and RF classifiers that use RPFB features are smaller than those that use fixed-window feature sets—with the same number of features. In this experiment, RF offers the best performance for both feature sets, with the error rate of 0.059 for $n = 70$ for the RPFB and 0.172 for $n = 100$ for the fixed-window feature sets. Similar to our observation for bearing dataset, one can see that the error rate of RPFB decays faster and with a relatively smaller number of features, e.g., $n = 12$, we can reach the error rate of 0.17 for the SVM classifier.

## 6 Conclusion

This paper introduced Random Projection Filter Bank (RPFB) as a simple and effective method for feature extraction from time series data. RPFB comes with a finite-sample error upper bound guarantee for a class of linear dynamical systems. We believe that RPFB should be a part of the toolbox for time series processing.

A future research direction is to better understand other dynamical system spaces, beyond the linear one considered here, and to design other variants of RPFB beyond those that are defined by stable linear autoregressive filters. Another direction is to investigate the behaviour of the discrepancy factor.

## A Auxiliary Results

We provide some auxiliary results that are used for the proofs in Section 4.

## A.1 Covering Number of a Ball in $\mathbb{R}^p$

The following lemma, quoted from van de Geer [2000], upper bounds the covering number of a ball with radius $B$ in $\mathbb{R}^d$.

**Lemma 9** (Covering number of a ball in an Euclidean space – Lemma 2.5 of van de Geer 2000). *A ball in $\mathbb{R}^d$ with radius $B$ w.r.t. Euclidean metric can be covered by $\left(\frac{4B+\varepsilon}{\varepsilon}\right)^d$ balls with radius $\varepsilon$.*

## A.2 Concentration Inequality for the Supremum of Martingalized Empirical Process

We quote a concentration inequality for a martingale variant of an empirical process from Rakhlin et al. [2014]. The definitions closely follows theirs.

Consider an arbitrary complete probability space $(\Omega, \mathcal{F}, P)$. Let $\mathcal{Z}$ be a separable metric space and $\mathcal{G} = \{g : \mathcal{Z} \to \mathbb{R}\}$ be a set of bounded real-valued functions on $\mathcal{Z}$. Consider a filtration $\mathcal{F}_1 \subseteq \mathcal{F}_2 \subseteq \mathcal{F}_3 \subseteq \cdots$ with $\cup_t \mathcal{F}_t \subseteq \mathcal{F}$. Let a sequence of random variables $Z_1, Z_2, \ldots$ be adapted to that filtration, i.e., $Z_t$ is $\mathcal{F}_t$-measurable.

Let $\mathbf{Rad}_T(G)$ be the sequential complexity of a function space $\mathcal{G}$ (Definition 3 of Rakhlin et al. 2014) and $\mathcal{N}_\infty(\varepsilon, \mathcal{G}, T)$ be its sequential covering number w.r.t. $\ell_\infty$-norm (Definition 4 of the same paper).

**Lemma 10** (Lemma 15 of Rakhlin et al. 2014—Slightly Simplified). *Let $\mathcal{G}$ be a $1$-bounded function space. Assume that $\mathbf{Rad}_T(\mathcal{G}) \geq \frac{1}{T}$ and $\mathcal{N}_\infty(\frac{1}{2}, \mathcal{G}, T) \geq 4$. Let $L = \max\{e^4, \sum_{j\geq 1}[\mathcal{N}_\infty(2^{-j}, \mathcal{G}, T)]^{-1}\}$. There exists an absolute constant $c$ such that for any $\varepsilon > 0$ and $T \geq 2$, we have*

$$\mathbb{P}\left\{ \left| \sup_{g \in \mathcal{G}} \frac{1}{T} \sum_{t=1}^{T} \left(g(Z_t) - \mathbb{E}\left[g(Z_t)|Z_{1:t-1}\right]\right) \right| > \varepsilon \right\} \leq 8L \exp\left( -\frac{\varepsilon^2}{c \ln^3(T) \mathbf{Rad}_T^2(\mathcal{G})} \right).$$

*The absolute constant $c$ can be selected to be $2^{17}$.*

*Remark* 7. The quantity $\sup_{g \in \mathcal{G}} \frac{1}{T} \sum_{t=1}^{T} \left(g(Z_t) - \mathbb{E}\left[g(Z_t)|Z_{1:t-1}\right]\right)$, which might be seen as a martingale variant of an empirical process, has been studied by van de Geer [2000] (Section 8.2). Theorem 8.13 there provides a concentration inequality in terms of a generalized entropy with bracketing in which the difference between functions are measured w.r.t. the Bernstein difference. It is interesting to see the relation between different complexity measures.

## A.3 Sequential Rademacher Complexity for the Regression Loss of a Ball in Hilbert Space

We provide an upper bound on the sequential Rademacher complexity for the squared regression loss with the function space being a bounded ball within a Hilbert space. This result is indeed very similar to Lemma 6 of Kuznetsov and Mohri [2015] (the extended version) with some minor differences. The first difference is that here we explicitly consider the truncated function space, whereas Kuznetsov and Mohri make the boundedness an assumption of their result. Additionally, their result considers a weighted version of the sequential Rademacher complexity, while we consider a uniform weight in this work. Since we do not need such a weighted version, we report a simplified result. Another minor difference is that they consider the $p$-th power of the regression error instead of the squared error of here.

**Proposition 11.** *Let $B, \Lambda > 0$. Consider a sequence $(X_1, X_2, \ldots)$ with $|X_t| \leq B$ (a.s.). Denote $Y_t = X_{t+1}$. Consider the function space*

$$\mathcal{G}_{\Lambda,B} \triangleq \left\{ g(x_{1:t}, y_{1:t}) = |\mathit{Tr}_B\left[\langle \alpha, \psi(\phi(x_{1:t})) \rangle\right] - y_t|^2 : \|\alpha\|_2 \leq \Lambda \right\}.$$

*Assume that $r \triangleq \sup_{w \in \mathbb{R}^n} \|\psi(w)\|_2 < \infty$. The sequential Rademacher complexity of this function space is upper bounded by*

$$\mathbf{Rad}_T(\mathcal{G}_{\Lambda,B}) \leq 32B \left( 1 + 4\sqrt{2} \ln^{3/2}(eT^2) \right) \frac{\Lambda r}{\sqrt{T}}.$$

*Proof.* We closely follow the proof of Lemma 6 of Kuznetsov and Mohri [2015] (the extended version). Define these function spaces:

$$\mathcal{F}_\Lambda \triangleq \left\{ (x_{1:t}, y_{1:t}) \mapsto \langle\, \alpha \,,\, \psi(\phi(x_{1:t})) \,\rangle - y_t \;:\; \|\alpha\|_2 \le \Lambda \right\},$$

$$\mathcal{F}_{\Lambda,B} \triangleq \left\{ (x_{1:t}, y_{1:t}) \mapsto \mathsf{Tr}_B\left[ \langle\, \alpha \,,\, \psi(\phi(x_{1:t})) \,\rangle \right] - y_t \;:\; \|\alpha\|_2 \le \Lambda \right\}.$$

We now use the structural properties of the sequential Rademacher complexity to relate the complexity of $\mathcal{G}_{\Lambda,B}$ to quantities that we can control easier. First note that the function $s \mapsto s^2$ with $s \in [-1,1]$ is 2-Lipschitz, so we use Lemma 13 and Proposition 14 of Rakhlin et al. [2014] to get

$$\mathbf{Rad}_T\left(\mathcal{G}_{\Lambda,B}\right) = \mathbf{Rad}_T\left( \left| 2B\left(\frac{\mathcal{F}_{\Lambda,B}}{2B}\right) \right|^2 \right) \le (2B)^2 \mathbf{Rad}_T\left( \left| \left(\frac{\mathcal{F}_{\Lambda,B}}{2B}\right) \right|^2 \right)$$

$$\le (2B)^2 8(2)\left( 1 + 4\sqrt{2}\ln^{3/2}(eT^2) \right) \mathbf{Rad}_T\left( \frac{\mathcal{F}_{\Lambda,B}}{2B} \right)$$

$$\le 32B\left( 1 + 4\sqrt{2}\ln^{3/2}(eT^2) \right) \mathbf{Rad}_T\left(\mathcal{F}_{\Lambda,B}\right). \tag{43}$$

Because $\mathcal{F}_{\Lambda,B} \subseteq \mathcal{F}_\Lambda$, by Proposition 14-1 of Rakhlin et al. [2014], we get that $\mathbf{Rad}_T(\mathcal{F}_{\Lambda,B}) \le \mathbf{Rad}_T(\mathcal{F}_\Lambda)$ too.

We have[10]

$$\mathbf{Rad}_T(\mathcal{F}_\Lambda) = \sup_{(x_{1:T}, y_{1:T})} \mathbb{E}_{\varepsilon_{1:T}} \left[ \sup_{\|\alpha\| \le \Lambda} \frac{1}{T} \sum_{t=1}^{T} \left( \langle\, \alpha \,,\, \psi\left(\phi(x_{1:t}(\varepsilon_{1:T}))\right) \,\rangle - y_t(\varepsilon_{1:T}) \right) \varepsilon_t \right]$$

$$= \sup_{(x_{1:T}, y_{1:T})} \mathbb{E}_{\varepsilon_{1:T}} \left[ \sup_{\|\alpha\| \le \Lambda} \frac{1}{T} \sum_{t=1}^{T} \varepsilon_t \langle\, \alpha \,,\, \psi\left(\phi(x_{1:t}(\varepsilon_{1:T}))\right) \,\rangle \right] +$$

$$\sup_{(x_{1:T}, y_{1:T})} \mathbb{E}_{\varepsilon_{1:T}} \left[ \sup_{\|\alpha\| \le \Lambda} \frac{1}{T} \sum_{t=1}^{T} y_t(\varepsilon_{1:T}) \varepsilon_t \right].$$

The second term on the RHS is zero because $\varepsilon_t$s are zero mean and $\varepsilon_t$ is independent of $y_t(\varepsilon_{1:T}) = y_t(\varepsilon_1, \dots, \varepsilon_{t-1})$.

Because $\sup_{\|u\|_2 \le 1} |\langle\, u \,,\, v \,\rangle| = \|v\|_2$, in a Hilbert space, we have

$$\sup_{(x_{1:T}, y_{1:T})} \mathbb{E}_{\varepsilon_{1:T}} \left[ \sup_{\|\alpha\| \le \Lambda} \frac{1}{T} \sum_{t=1}^{T} \varepsilon_t \langle\, \alpha \,,\, \psi\left(\phi(x_{1:t}(\varepsilon_{1:T}))\right) \,\rangle \right] =$$

$$\sup_{(x_{1:T}, y_{1:T})} \mathbb{E}_{\varepsilon_{1:T}} \left[ \frac{\Lambda}{T} \left\| \sum_{t=1}^{T} \varepsilon_t \psi\left(\phi(x_{1:t}(\varepsilon_{1:T}))\right) \right\|_{\mathcal{H}_0} \right] \le$$

$$\frac{\Lambda}{T} \sup_{(x_{1:T}, y_{1:T})} \sqrt{ \mathbb{E}_{\varepsilon_{1:T}} \left[ \left\| \sum_{t=1}^{T} \varepsilon_t \psi\left(\phi(x_{1:t}(\varepsilon_{1:T}))\right) \right\|_{\mathcal{H}_0}^2 \right] } =$$

$$\frac{\Lambda}{T} \sup_{(x_{1:T}, y_{1:T})} \sqrt{ \mathbb{E}_{\varepsilon_{1:T}} \left[ \sum_{s,t=1}^{T} \varepsilon_t \varepsilon_s \psi\left(\phi(x_{1:t}(\varepsilon_{1:T}))\right) \psi\left(\phi(x_{1:s}(\varepsilon_{1:T}))\right) \right] } =$$

$$\frac{\Lambda}{T} \sup_{(x_{1:T}, y_{1:T})} \sqrt{ \mathbb{E}_{\varepsilon_{1:T}} \left[ \sum_{t=1}^{T} \left\| \psi\left(\phi(x_{1:t}(\varepsilon_{1:T}))\right) \right\|_{\mathcal{H}_0}^2 \right] } \le \frac{\Lambda r}{\sqrt{T}},$$

where in the last inequality we used the assumption that for any $w$, $\|\psi(w)\|_{\mathcal{H}_0} \le r$. Together with (43), we reach the desired upper bound.

$\square$

*Remark* 8. Comparing this result with Lemma 6 of Kuznetsov and Mohri [2015] (the extended version), one may notice that the definitions of $r$ are slightly different even though both $r$ appear linearly in the upper bound. Here we have $r = \sup_w \|\psi(w)\|_2$, while they have $r' = \sup_w \kappa(w, w)$, with $\kappa$ being the kernel of an RKHS corresponding to the feature map $\psi$. By definition, $\kappa(w, w) = \langle \psi(w), \psi(w) \rangle_{\mathcal{H}_0} = \|\psi(w)\|_{\mathcal{H}_0}^2$, so their choice of $r'$ corresponds to $r^2$. This appears to be a typo.

The next proposition is useful to verify a technical condition on the covering number in Lemma 10.

**Proposition 12.** *Consider the function space* $\mathcal{G}_{\Lambda,B}$ *(36). Assume that the sequence* $X_1, X_2, \cdots$ *is such that* $|X_t| \leq B$ *(a.s.). Denote* $Y_t = X_{t+1}$. *The sequential covering number satisfies*

$$\mathcal{N}_\infty(\varepsilon, \mathcal{G}_{\Lambda,B}, T) \geq \frac{2B^2 \Lambda r}{\varepsilon}.$$

*Proof.* Note that each $g(x_{1:t}, y) \in \mathcal{G}_{\Lambda,B}$ is in the form of

$$g(x_{1:t}, y_{1:t}) = |\mathsf{Tr}_B\left[\langle \alpha, \psi(\phi(x_{1:t})) \rangle\right] - y_t|^2 = \left|\tilde{h}(w_t) - y_t\right|^2.$$

for $\tilde{h}(w_t) \in \tilde{\mathcal{H}}_{\Lambda,B}$ (26). To provide a lower bound on the sequential covering number, let us pick $g_1, g_2 \in \mathcal{G}_{\Lambda,B}$ and their corresponding $\tilde{h}_1, \tilde{h}_2 \in \tilde{\mathcal{H}}_{\Lambda,B}$. Note that the worst case distance, over the choice of $w_{1:T}, y_{1:T}$, appearing in the definition of the sequential covering number can be lower bounded by

$$\sup_{w_{1:T}, y_{1:T}} \max_{1 \leq t \leq T} |g_1(w_t(\varepsilon), y_t(\varepsilon)) - g_2(w_t(\varepsilon), y_t(\varepsilon))|^2 =$$
$$\sup_{w_{1:T}, y_{1:T}} \max_{1 \leq t \leq T} |h_1(w_t(\varepsilon)) - h_2(w_t(\varepsilon))| \times |h_1(w_t(\varepsilon)) + h_2(w_t(\varepsilon)) - 2y_t(\varepsilon)| \geq$$
$$2B \sup_{w_{1:T}} \max_{1 \leq t \leq T} |h_1(w_t(\varepsilon)) - h_2(w_t(\varepsilon))|,$$

because we are free to choose the sequence $y_{1:T}$ only with the constraint that $|y_t| \leq B$, so the second multiplicative term would be at least as large as $2B$. Therefore, if we need more than $N_\varepsilon$ functions to $\varepsilon$-cover $\tilde{\mathcal{H}}_{\Lambda,B}$, we need more than $N_\varepsilon$ functions to $2B\varepsilon$-cover $\mathcal{G}_{\Lambda,B}$ as well.

According to the discussion after Lemma 15 of Rakhlin et al. [2014], the class of linear functions $\tilde{\mathcal{H}}_{\Lambda,B}$ with $r = \sup_w \|\Psi(w)\|_2$ has a lower bound on the covering number of $\mathcal{N}_\infty(\varepsilon', \tilde{\mathcal{H}}_{\Lambda,B}T) \geq \frac{B\Lambda r}{\varepsilon'}$ (we use a linear scaling of the space to get this from their result). So

$$\mathcal{N}_\infty(\varepsilon, \tilde{\mathcal{G}}_{\Lambda,B}T) \geq \frac{2B^2 \Lambda r}{\varepsilon}.$$

$\square$

# B Detail of Experiments

## B.1 Software

All experiments were implemented in Python. We used SciPy [Jones et al., 2001–] to implement the filters in RPFB. We used scikit-learn [Pedregosa et al., 2011] to implement various estimators, and Keras package Chollet [2015] on top of Theano [Theano Development Team, 2016] to implement LSTM.

## B.2 Hardware

Experiments of Section 5.2 were run on a machine with Intel Xeon CPU E5-2620 v4 @ 2.10GHz and 128GB of RAM. We used NVIDIA Titan X (Pascal) (12GB) for the GPU computation of the LSTM experiment. We did not optimize the code to maximally benefit from the hardware.

### B.3 Model Selection for LR and SVM for Bearing Dataset

For each setting (number of features, RPFB vs. Window-based) and for each independent run of RPFB, we perform a model selection to choose the best hyper-parameters by doing a 3-split time series cross-validation over the training dataset.

For LR, the model selection is over the regularization coefficient (10 parameters over the logarithmic grid between $10^{-3}$ and $10^{+3}$). For SVM, the model selection is over both the regularization coefficient and the Gaussian kernel scale parameters. This is done over a $15 \times 15 = 225$ logarithmic grid.

For RF, we do not perform any model selection on the number of trees; it is always set to 25 in the reported results.

### B.4 LSTM for Bearing Dataset

We use Adam as the optimizer [Kingma and Ba, 2015], with parameters suggested in their Algorithm 1: the stepsize $\alpha = 0.001$, the decay rates for the moment estimates $\beta_1 = 0.9$ and $\beta_2 = 0.999$, and $\varepsilon = 10^{-8}$, in their notations. We use a mini-batch of size 128. We train for 50 epochs over the training set. We have not performed any hyper-parameter optimization. For each setting of the parameters and architectures, we run the experiment 20 times to measure the variability of the results.

### B.5 Distribution of RPFB for Bearing Dataset

For the bearing experiment, we use a slightly different sampling distribution for the poles of RPFB. We choose a fraction ($\frac{1}{4}$) of the filters to have poles on the unit circle, i.e., $Z'_j = e^{j\theta}$ and $\bar{Z}'_j = e^{-j\theta}$. The rest, as before are either on the real line, or are $Z'_j = re^{j\theta}$ and $\bar{Z}'_j = re^{-j\theta}$ with $r < 1$.

#### Acknowledgments

We would like to thank the anonymous reviewers for their helpful feedback.

## Footnotes

[1]We assume that $A$ and $B$ both have roots within the unit circle, i.e., they are stable.

[2]The fact that both of these polynomials have a leading term of 1 does not matter in this argument.

[3]For continuous-time systems, we may use Laplace transform instead of Z-transform, and have similar representations.

[4]One could generate different types of filters, for example those with nonlinearities, but in this work we focus on linear AR filters to simplify the analysis.

[5]Evidently, there might be a better minimizer within a larger function space $\tilde{\mathcal{F}}_{\Lambda'}$ with $\Lambda' > \Lambda$.

[6]Our definition is a simplified version of the original one (by selecting $q_t = 1/T$ in the original paper) and is adapted to our choice of function space.

[7]So the distribution of poles is not uniform over the unit circle.

[8]Available from http://www.mfpt.org/faultdata/faultdata.htm.

[9]Available from http://ecg.mit.edu/time-series/.

[10]We use the notation of Rakhlin et al. [2014] in defining a function specified by a path on a tree. The notation $x_{1:t}(\varepsilon_{1:T})$ should be understood similarly: It is a sequence $(x_1(\emptyset), x_2(\varepsilon_1), \dots, x_t(\varepsilon_{1:t-1}))$.