[Reviews · NeurIPS 2017]

Reviewer 1



This paper proposes a new algorithm for learning on time series data. The basic idea is to introduce random projection filter bank for extracting features by convolution with the time series data. Learning on time series data can then be performed by applying conventional machine learning algorithms to these features. Theoretical analysis is performed by studying both the approximation power and statistical guarantees of the introduced model. Experimental results are also presented to show its effectiveness in extracting features from time series. Major comments: (1) It seems that the algorithm requires the information on both $M$ and $\epsilon_0$, which depend on the unknown target function $h^*$ to be estimated. Does this requirement on an apriori information restrict its practical application? Since $Z_i'$ are complex numbers, the hypothesis function in (9) would take complex outputs. It is not clear to me how this complex output is used for prediction. (2) It is claimed that the discrepancy would be zero if the stochastic process is stationary. However, this seems not to be true as far as I can see. Indeed, the discrepancy in Definition 1 is a random variable even if the stochastic process is i.i.d.. So, it depends on the realization of $X$. If this is true, Theorem 1 is not clear to understand since it involves a discrepancy which would not be zero in the i.i.d. learning case. (3) There exists some inconsistences in Algorithm 1 and materials in Section 4. Algorithm 1 is a regularization scheme while ERM is considered in Section 4. There are $m$ time series in Algorithm 1, while $m=1$ is considered in Section 4. The notation is not consistent, i.e., $f$ is used in Algorithm 1 and $h$ is used in Section 4. (4) The motivation in defining features in Section 3 is not clear to me. It would be better if the authors can explain it more clearly so that readers without a background in complex analysis can understand it well. Minor comments: (1) line 28: RKSH (2) line 65, 66: the range of $f\in\mathcal{F}$ should be the same as the first domain of $l$ (3) eq (11), the function space in the brace seems not correct (4) the notation * in Algorithm 1 is not defined as far as I can see (5) $\delta_2$ should be $\delta$ in Theorem 1 (6) line 254: perhaps a better notation for feature size is $n$? (7) Appendix, second line in equation below line 444, a factor of $2$ is missed there? (8) Appendix, eq (28), the identity should be an inequality? I suggest the authors to pay more attention to (28) since the expectation minus its empirical average is not the same as the empirical average minus expectation.

Reviewer 2



The paper develops a technique for extracting features from sequential time series data analogous to (in a sense, generalizing) ideas from random projections for dimensionality reduction and random features for kernel space approximation. At a high level view, the algorithm produces a set of (randomized) feature maps by projecting the input sequence onto randomly generated dynamical filters, which in turn are argued to approximate a certain class of dynamical filters. I unfortunately lack the fluency in dynamical systems needed to evaluate the mathematical novelty/significance and the correctness of the approach. I had to take several of the paper's claims at face value. From the point of view of statistical learning, and as a possible user, I find the overall idea intriguing and I found the paper's high-level organization clear. A few suggestions for improvement, or possible concerns, primarily bearing on novelty/significance for a machine learning audience: 1. The experiments are described without background. There is a missing mention of scale, and whether these tasks are considered challenging (or are closer to being small-scale benchmarks), and this background seems more crucial for the wider NIPS audience. The introducing presents the low-computational-power setting as motivation for this work, so to what extent do these datasets represent tasks relevant to a computationally constrained setting? Also, an interpretation of the results is missing, which takes away from the significance signal. 2. It seems worth discussing (or evaluating?) alternatives in the wider context of machine learning today. This paper argues that it provides a good data-independent representation when RNNs are too compute-intensive (line 39). Concretely, what justifies this claim? And when computational resources are not a tight concern, are the two approaches at all in the same task performance ballpark? One could optimistically hope for a positive result that provides a decent baseline for consideration when training RNNs. Minor/specific: - Line 21: "convoluted" -> "convolved"? - Line 39: "power methods" -> "powerful methods"? - Line 40: why might RNNs be computationally infeasible? Any example? - Line 62: why have $Y_{i,T_{i}} = +1$? - Line 114: "prune" -> "prone"?

Reviewer 3



*** Following authors feedback that addressed two of my main concerns about experiments and theory I am updating my rating *** This paper proposes a feature extraction mechanism that applies an ensemble of order one autoregressive filters whose poles are randomly sampled from the unit circle. The authors show that such filters span a higher order filter. These features can be plugged into a conventional machine learning algorithm to construct a predictive model from a time series to a target variable. The authors provide a finite sample error bound. The paper is well written. The idea to adapt random projection methods to time series by using random filters is interesting and the method outperforms fixed window based features. However, to demonstrate the usefulness of this approach, the proposed method should be compared to other methods that can automatically learn how to integrate the history of signal. A natural family of methods to compare, are those based on state space representations, such as simple RNNs or the Statistical Recurrent Unit [Oliva etal. 2017]. The suggested method applies to univariate signals because the generated filter bank is univariate. It would be interesting to pursue how does this approach scale up to multi-dimensional signals, to increase the significance of this paper. The error bound requires some clarifications: (1) Does it apply to case 2 as well as case 1? (2) n appears in the numerator of the first term as well as in the denominator of the second one, which may reflect a bias-variance tradeoff in the choice of number of filters. This trade-off implies that an optimal choice of n depends on T^(1/3). If so, how does the method handle multiple time series with varying lengths? (3) Following (2), is the number of filters a hyperparameter that should be tuned by cross validation? In the experimental section, is the task is to classify entire sequences or to monitor the rotating system at every time point? In case it’s the entire sequence, how is it done in practice, do you consider the label of the last observation Y_t as the label of the entire series, or do you exploit feature associated with earlier time points? Does the dataset include three time series or three sets of time series? In each case, how was it split to train and test? To summarize, this is an interesting manuscript, but more work is needed to mature the proposed methodology.